# The exquisitely preserved integument of *Psittacosaurus* and the scaly skin of ceratopsian dinosaurs

Phil R. Bell [1,6✉], Christophe Hendrickx [2,6✉], Michael Pittman [3,4✉], Thomas G. Kaye [4] & Gerald Mayr[5]

The Frankfurt specimen of the early-branching ceratopsian dinosaur *Psittacosaurus* is remarkable for the exquisite preservation of squamous (scaly) skin and other soft tissues that cover almost its entire body. New observations under Laser-Stimulated Fluorescence (LSF) reveal the complexity of the squamous skin of *Psittacosaurus*, including several unique features and details of newly detected and previously-described integumentary structures. Variations in the scaly skin are found to be strongly regionalized in *Psittacosaurus*. For example, feature scales consist of truncated cone-shaped scales on the shoulder, but form a longitudinal row of quadrangular scales on the tail. Re-examined through LSF, the cloaca of *Psittacosaurus* has a longitudinal opening, or vent; a condition that it shares only with crocodylians. This implies that the cloaca may have had crocodylian-like internal anatomy, including a single, ventrally-positioned copulatory organ. Combined with these new integumentary data, a comprehensive review of integument in ceratopsian dinosaurs reveals that scalation was generally conservative in ceratopsians and typically consisted of large subcircular-to-polygonal feature scales surrounded by a network of smaller non-overlapping polygonal basement scales. This study highlights the importance of combining exceptional specimens with modern imaging techniques, which are helping to redefine the perceived complexity of squamation in ceratopsians and other dinosaurs.

[1] School of Environmental and Rural Science, University of New England, Armidale, NSW 2351, Australia. [2] Unidad Ejecutora Lillo, CONICET-Fundación Miguel Lillo, Miguel Lillo 251, 4000 San Miguel de Tucumán, Tucumán, Argentina. [3] School of Life Sciences, The Chinese University of Hong Kong, Shatin, Hong Kong SAR, China. [4] Foundation for Scientific Advancement, Sierra Vista, AZ, USA. [5] Ornithological Section, Senckenberg Research Institute and Natural History Museum Frankfurt, Senckenberganlage 25, D-60325 Frankfurt am Main, Germany. [6]These authors contributed equally: Phil R. Bell, Christophe Hendrickx. ✉email: pbell23@une.edu.au; christophendrickx@gmail.com; mpittman@cuhk.edu.hk

The first report of scaly skin in a non-avian dinosaur (hereafter, dinosaur) was that of a sauropod by Mantell in 1852, but which was incorrectly identified as a giant crocodylian at the time[1–4]. The discovery of "typical" reptilian scales among dinosaurs for the remainder of the nineteenth and much of the twentieth century has been regarded with some degree of ambivalence[5,6], although the discovery of feathered specimens from Liaoning Province of China in the 1990s (e.g. refs. [7,8]) has since spurred an intense interest in the integument of dinosaurs. However, outside Hadrosauridae, which includes several "mummified" specimens covered with skin (see review by refs. [9,10]), the scaly integument of dinosaurs is still surprisingly poorly known. This is particularly the case for marginocephalians, in which the skin has been reported in only six ceratopsian taxa, namely, *Centrosaurus*[11,12], *Chasmosaurus*[13], *Nasutoceratops*[14], *Protoceratops*[15], *Psittacosaurus*[16–19], and *Triceratops*[20]. Aside from *Psittacosaurus*, preserved skin in other ceratopsians is restricted to Coronosauria (i.e., Protoceratopsidae + Ceratopsoidea) and limited in body coverage. Scalation in Ceratopsia frequently consists of large rounded feature scales surrounded by smaller polygonal basement scales (e.g. refs. [11,13,18,21,22]), but despite such generalizations, ceratopsians also show diverse skin morphologies with recognizable interspecific differences in the architecture of both feature and basement scales[14].

The famous Frankfurt specimen of *Psittacosaurus* sp. deposited in the Senckenberg Research Institute and Natural History Museum, Germany (SMF R 4970) is endowed with one of the most complete coverings of squamous skin in any dinosaur and is one of the few non-hadrosaurid ornithischians with integument covering a large portion of the body (Fig. 1). It is also the sole marginocephalian to include skin from the region of the head, limbs and tail and the only dinosaur in which the keratinous cranial horn is preserved[23] (see "skin morphology in ceratopsian dinosaurs" below). More importantly, the integument of *Psittacosaurus* SMF R 4970 preserves evidence of color patterns and countershading and is the only dinosaur to preserve an umbilical scar[24] or the cloaca[19,25]. The latter was recently revealed to be unique among tetrapods in having a V-shaped convergence of the two darkly-pigmented lateral lips, in addition to a bulbous dorsal lobe[25].

*Psittacosaurus* integument was first reported in a subadult individual of *P. mongoliensis* (AMNH FARB 6260; see the list of institutional abbreviations in Supplementary Note 1) by Sereno[26] (p. 248) on the plantar surface of the right pes (metatarsals I-IV). The reticulate scales were described as minute (<1 mm), rounded tubercules that did not form any pattern[26]. Ji[27] was, however, the first to formally describe scales in *Psittacosaurus*, which consisted of small (<3–4 mm in diameter) polygonal and triangular scales, near the left humerus. Five years later, the same author described

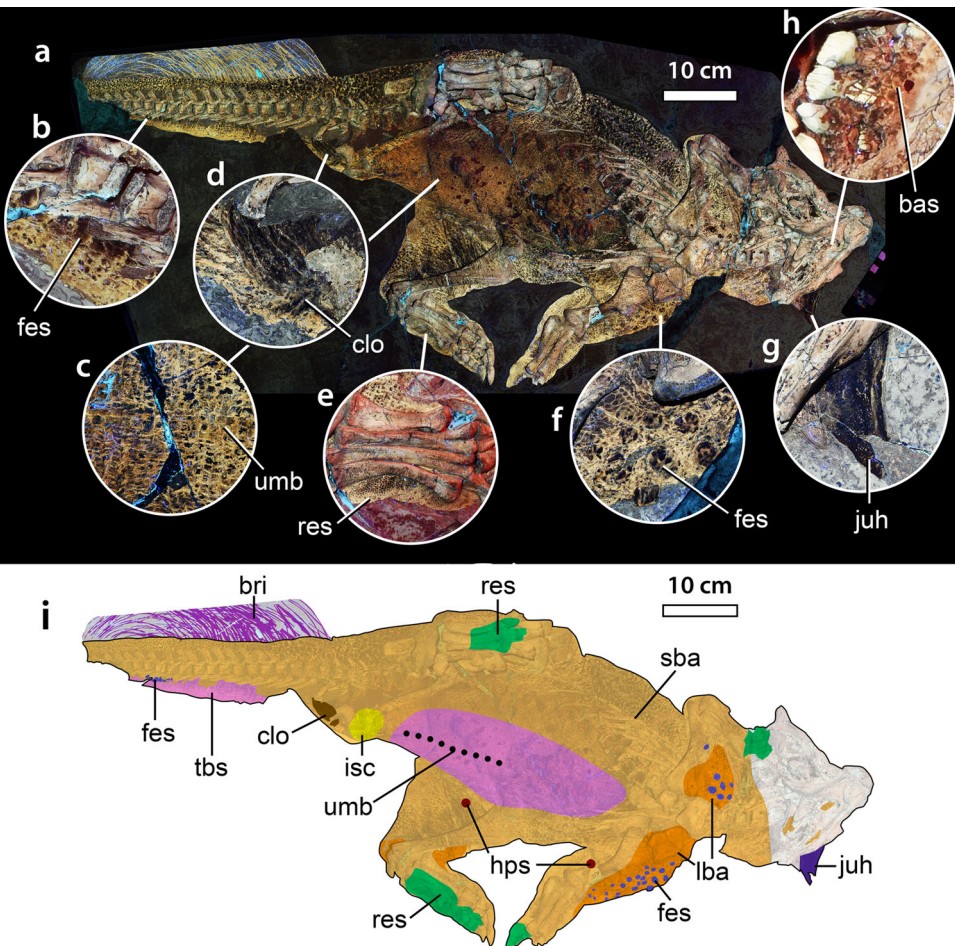

**Fig. 1 *Psittacosaurus* sp. (SMF R 4970) under laser-stimulated fluorescence (LSF) and distribution of different integumentary types. a** Specimen under LSF, with close ups on different integumentary types (**b–h**). **b** Feature scales of the ventral portion of the tail. **c** Umbilical scar with transverse rows of quadrangular scales on the abdomen. **d** Cloaca. **e** Reticulate scales of the left foot. **f** Feature scales of the shoulder. **g** Jugal horn. **h** Basement scales of the mandible. **i** Schematic showing the distribution of different integumentary types. bas basement scale, bri tail bristles, clo cloaca, fes feature scale, hps hexagram pattern of basement scales, ish ischial callosity, juh jugal horn, lba large basement scales, res reticulate scales, sba small basement scales, tbs transversely banded scales, umb umbilical scar.

several patches of skin of a specimen from western Liaoning probably representing a psittacosaurid[21]. Mayr et al.[16] provided a brief account of the skin morphology in *Psittacosaurus* based on the exquisitely preserved specimen SMF R 4970, focusing their attention on the bristle-like integumentary structures of the tail. The latter received a more thorough treatment by Mayr et al.[23] who argued that the "bristles" were homologous to the mono-filaments of theropods such as *Beipiaosaurus*. Lingham-Soliar and Plodowski[18] also described the scale pattern and distribution in *Psittacosaurus* in more detail, listing three types of scales in SMF R 4970, i.e., large, rounded plate-like scales, smaller polygonal scales or tubercles, and round pebble-like scales. Nevertheless, Lingham-Soliar and Plodowski[18] only superficially described each scale morphotype and did not provide information on the scale pattern and morphology in the manus, pes and cloaca region. Those authors, however, revealed light and dark cryptic patterns created by the association of the tubercles and plate-like scales which were described as the first evidence of countershading in a dinosaur (see also ref. [19]).

In recent years, laser-stimulated fluorescence (LSF) has become a powerful tool in paleontology for highlighting and/or revealing additional soft tissue details in fossils that are otherwise unseen under white light conditions[28,29]. The application of this technology to *Psittacosaurus* SMF R 4970 has quite literally illuminated new aspects of the tail bristles[23] and color patterns[19] and permitted the identification of the cloaca[25] and umbilical scar[24]. In addition, Vinther et al.[19] (their supplementary information) provided some preliminary observations of the scale architecture and taphonomy of SMF R 4970 using this technique. The purpose of this study is to augment these earlier descriptions with strict attention to the scale architecture with the aim of providing an even clearer picture of the appearance and palaeobiology of one of the most well-preserved dinosaurs in existence. We use this to inform a detailed review of skin morphology and distribution across ceratopsians with an aim of better understanding squamation patterns across Dinosauria.

## Results and discussion

**Head and neck.** Integument on the head and neck is the least well preserved in SMF R 4970 (Fig. 2). Discontinuous patches of darkly-pigmented skin are present on various parts of the underside of the skull with individual basement scales best discerned on the medial surface of the left mandible, the lateral surface of the right mandible, the posterior extremity of the palate and the right jugal (Fig. 2c–f). The integument that covers the left mandible and the palate likely corresponds to the skin formerly covering the throat region between the lower jaws. Scale shapes are often difficult to distinguish but the patches on the palate (Fig. 2e) and the medial side of the left mandible show oval to subcircular basement scales (Fig. 2d, f). The smallest scales are found on the palate where they range between 0.5–1 mm in diameter whereas those from the two patches on the medial side of the mandible, next to the dentition and on the distal portion of the ramus, are 1–3 mm in diameter.

The most conspicuous feature of the cranial integument is a darkly-pigmented triangle of soft tissue on the left side of the skull interpreted as the keratinous "sheath" of the jugal horn[19,23] (Fig. 2a–c, i). Although clearly associated with the osseous jugal horn, the "sheath" is offset anterior to and projects at least 23 mm laterally beyond the tip of the jugal horn (Fig. 2c). A break in the rock truncates the anterolateral edge of the "sheath", which lies at a deeper level in the rock matrix than the bone itself indicating the "sheath" was anatomically dorsal to the bony core. The surface of the "sheath" is uniform to somewhat granular or mottled in appearance and devoid of scales. The ventral surface of the right osseous jugal horn, however, preserves darkly-colored globular and branching structures that appear to represent the pigmented interstitial tissue between non-pigmented epidermal scales (Fig. 2i). The scales themselves are polygonal (hexagonal?), 3–4 mm in diameter. On the ventral surface of the jugal horn on the left side, however, individual scales are not discernible, only indistinct dark mottling.

Skin and scales are well preserved on either side of the cervical vertebrae delimiting a thick neck, which tapers from the back of

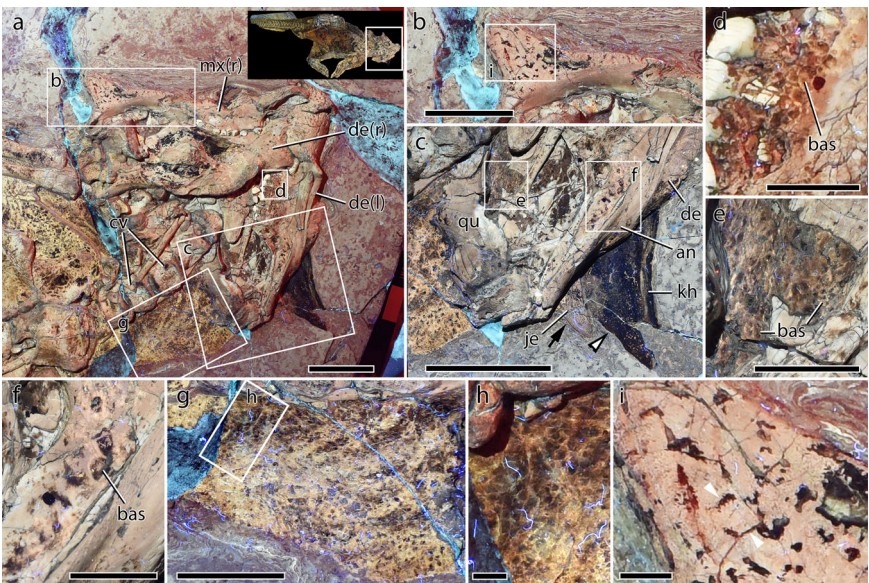

**Fig. 2 Integument of the head and neck of *Psittacosaurus* SMF R 4970 under LSF. a** Head and neck with inset showing the region depicted in **a**. **b** Close up of the right jugal horn. **c** Close up of the left jugal horn and dark region interpreted as the keratinous "horn". Note the offset between the posterior edges of the jugal horn (black arrow) and the keratinous "horn" (black and white arrowhead). Close up on the integument of the head, from **d** the dorsomedial side of the left ramus of the mandible, directly ventral to the lower dentition; **e** the palatal region; and **f** the ventromedial side of the distal portion of the left ramus of the mandible. **g** Close up of the integument on the left side of the cervical vertebrae. **h** Detail of the squamous integument shown in **f**. **i** Detail of the right jugal horn (boxed region in **b**) with pigmented strips forming polygons (arrowheads). an angular, cv cervical vertebrae, de dentary, je jugal horn, kh keratinous "horn", mx maxilla, qu quadrate. Scale bars equal 5 cm (**a**, **c**), 3 cm (**b**, **d**), and 5 mm (**e**, **f**).

the skull to a minimum width equal to approximately three times the width of the cervical vertebrae (as preserved in ventrolateral view) in line with the third post-axial vertebra (Fig. 2a). Scales on the neck form a basement of typically anteroposteriorly elongated ($\bar{x}$ length = 1.6 mm; Supplementary Table 1), lenticular-to-weakly polygonal scales, and transitioning posteriorly to more elliptical or rounded-polygonal (i.e., less elongated) scales in the vicinity of the third and fourth post-axial vertebrae (Fig. 2g–h).

**Shoulder and forelimbs**. The soft tissue outline describes a forelimb that was robust, stocky and almost columnar, the greatest measurable width occurring at the mid-length of the humerus (4.5 times wider than the minimum width of the humerus), tapering distally with a modest constriction associated with the inner elbow (Fig. 3a). The pectoral girdle and forelimb are covered in a basement of typically small ($\bar{x}$ diameter = 1.9 mm), non-overlapping polygonal or rounded-polygonal (3–6 sided) scales. Interstitial tissue is relatively wide in the region of the coracoid (~1 mm wide) but elsewhere the scales closely abut one another. The arrangement of the scales differs somewhat along the forelimb: immediately anterior to the humerus, scales are more proximodistally elongate (~2 × 1 mm) and form columns parallel to the long axis of the humerus. On the distal humerus and the area posterior to the mid-shaft, the scales form patterns consisting of a central polygonal scale ($\bar{x}$ diameter = 1.2 mm) surrounded by five or six small triangular elements ($\bar{x}$ length = 0.4 mm). This arrangement recalls a hexa-gram, or 6-pointed star (Fig. 3c). Each "star" closely abuts its neighbor so that the pattern is continuous across this region. This unusual pattern is also present on the hindlimb (see below) and has been described from the proximal humerus of *Nasutocera-tops*, where the "stars" are comparatively large (central scale diameter up to 11 mm[14]). Elsewhere on the forelimb of SMF R 4970, the basement scales are less regularly arranged but each

scale is always surrounded by six of its neighbors. Set within this basement are relatively large feature scales disposed along the presumed anterior–anteromedial surface of the girdle and fore-limb between the coracoid to a point in line with the mid-length of the humerus (Fig. 3b). The feature scales themselves are almost cylindrical or truncated-cone shaped, with a circular basal cross-section ($\bar{x}$ diameter = 8.8 mm) and a height of up to 6.8 mm (Fig. 3d). The surfaces appear smooth but are pigmented by five or six broad dark stripes that extend from the scale base, con-verging apically but terminating to form an unpigmented star-shaped pattern at the scale apex (Fig. 3d). Feature scales are spaced ~6–9 mm apart along two or three roughly-formed ver-tical rows (parallel to the long axis of the humerus) ~5–6 mm apart. The basement scales immediately surrounding the feature scales do not differ in size or arrangement from the remaining basement scales (i.e., they do not form a rosette pattern sensu Pittman et al.[4]). Whether feature scales continued onto the lateral surface of the forelimb is unknown. The pattern of small (~2 mm) polygonal basement scales continues distally on the forelimb with the exception of the inner elbow where they are even smaller ($\bar{x}$ diameter = 1.1 mm). At the junction between the distal ante-brachium and the manus, the scale covering abruptly transitions to tiny reticulate scales ($\bar{x}$ diameter = 0.5 mm) on the palmar surface of the manus. Although incomplete, the contour of the soft tissues surrounding the manus indicate the presence of a fleshy "heel" (Fig. 3e).

**Trunk and abdomen**. The integument covering the trunk can be divided into (and differs between) two broad regions: the soft abdomen between the rib cage and extending posteriorly to the pelvic girdle, and the lateral flanks associated with the ribs. Despite extensive integument on the right flank (i.e., lateral to the rib cage), individual scales are not easily identified due to the heavy pattern of pigmentation. Individual scales from the flanks

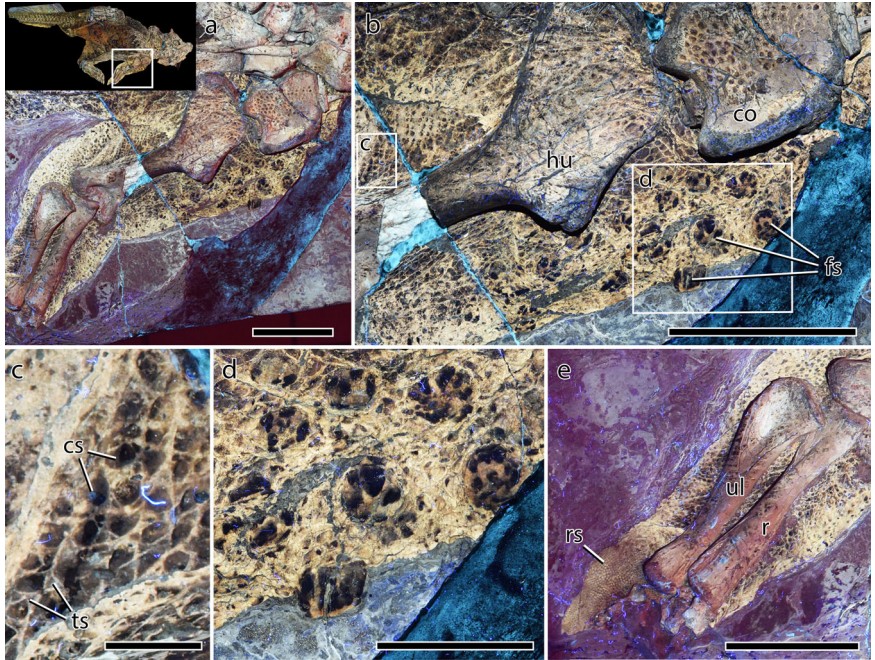

**Fig. 3 Integument of the forelimb and pectoral region of *Psittacosaurus* SMF R 4970 under LSF. a** Left forelimb in medial aspect with inset showing the region depicted in **a**. **b** Shoulder region showing large basement scales on the coracoid, grid-like arrangement of smaller basement scales dorsal to the humerus, and large pigmented feature scales on the anterior brachium. **c** Detail of boxed region in **b** showing hexagram arrangement of basement scales. **d** Detail of the raised feature scales close to the shoulder joint showing striped pigmentation. **e** Antebrachium and fleshy palmar pad bearing reticulate scales. co coracoid, cs central scale within hexagram pattern, fes feature scales, hu humerus, r radius, res reticulate scales, ts triangular scales within hexagram pattern, ul ulna. Scale bars equal 5 cm (**a**, **b**, **e**), 2 cm (**d**), and 5 mm (**c**).

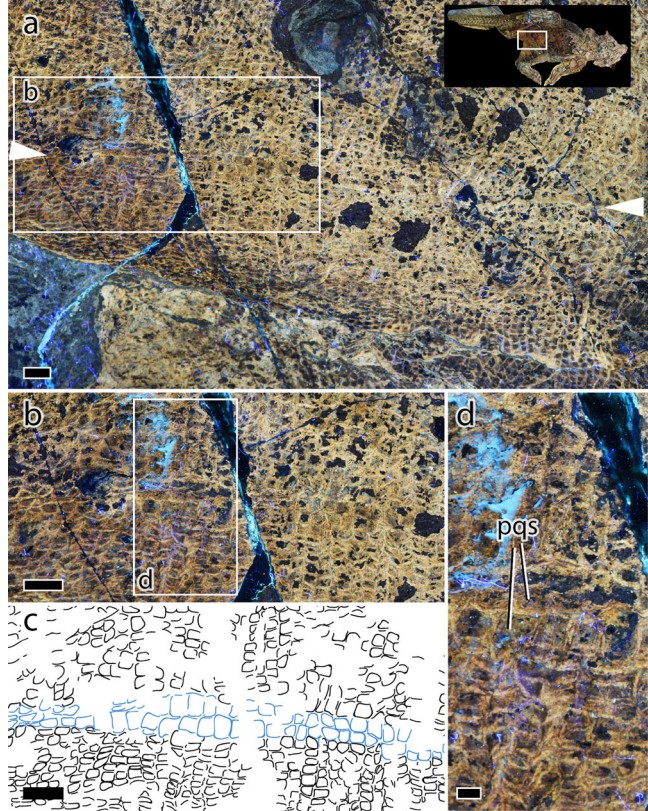

**Fig. 4 Abdominal skin in *Psittacosaurus* SMF R 4970 under LSF.**
**a** Abdominal region with inset showing the region depicted in **a**. The maximal anteroposterior extent of the paired scales is indicated by arrowheads. **b**, **c** Close up of boxed region in **a** showing paired quadrangular scales (blue outline in **c**). Transverse banding is visible in the remaining abdominal scales (black outlines in **c**). **d** Close up of paired quadrangular scales (pqs). Scale bars equal 5 mm (**a**–**c**), and 2 mm (**d**).

are better seen on the left side where they are preserved medial and lateral to the thoracic ribs. Basement scales in this region are anteroposteriorly-elongated and roughly diamond-shaped ($\bar{x} = 2.3 \times 1.7$ mm). A small number of larger circular-to-irregular feature scales (3–4 mm diameter) are interspersed throughout these diamond-shaped basement scales although not enough are discernible to identify any clear pattern in the arrangement of the feature scales. The latter are smaller, do not appear to be significantly elevated, and are uniformly dark compared to the striped, cylindrical feature scales in the shoulder region. The second region, corresponding to the soft underparts of the animal, between the rib cage and extending posteriorly between the ischia, is covered by quadrangular scales arranged into distinct transverse "bands" as is typical of modern crocodylians and some squamates (e.g., *Uromastix*; pers. obs.; Fig. 4; see Supplementary Data). Scales are small ($\bar{x}$ length = 1.4 mm), becoming slightly larger ($\bar{x}$ length = 1.8 mm) and less distinctly banded in the anterior part of the abdomen (i.e., anterior of the presumed gastric mill). Extending anteriorly from the ischial callosity (see "tail and cloaca" below), the transverse bands are broken along the ventral mid-line of the animal by a distinct longitudinal row of paired quadrangular scales ($\bar{x}$ length = 2.5 mm) (Fig. 4b–d). This mid-line row extends from just in front of the ischial symphysis anteriorly for ~13 cm.

**Hindlimbs**. The soft tissues surrounding the hind limb show a remarkably broad (anteroposteriorly) crus (i.e., part of the leg

between the knee and the foot). As preserved, the soft tissues are broadest around the left knee (~3 times the anteroposterior length of the proximal tibia) tapering distally to the ankle joint at which point the flesh more closely shrouds the foot bones (Fig. 5a). The bulk of this tissue is posterior to the tibia, presumably corresponding to a large area of powerful leg retractor muscles. The tibia itself is close to and nearly parallel to the anterior edge of the integumentary outline. Although the position of the animal obscures both femora, the volume of flesh shrouding the crus makes it unlikely that the upper leg was well separated from the body; there is no indication of a crease in the flesh at the back of the knee. Instead, the web of tissue extending proximally from the ankle joint probably bounds the upper leg closely to the body as in many extant quadrupedal mammals (e.g., *Equus, Bos*).

Scales on the lower leg are hard to discern due to the dark color of the fossil integument. In the region posterior to the tibia, close to where the leg meets the torso, scales occasionally form a hexagram pattern consisting of a central subcircular scale ($\bar{x}$ diameter = 1.3 mm) surrounded by a number of smaller triangular scales ($\bar{x}$ length—0.5 mm) similar to those on the forelimb (Fig. 5d, e). The precise number of triangular scales cannot be determined on the hindlimb, but these number at least five on the best-preserved scales, suggesting at least six scales completed each "star". In some cases, the central scale is darkly pigmented, whereas the triangular scales are lighter colored. It is unclear whether this hexagram pattern continues more distally on the crus as the dense pigmentation, or lack thereof, makes individual scales less conspicuous; dark spots visible on other parts of the hindlimb probably pertain to pigmented subcircular scales although the margins of the scales themselves are not always visible. Similarly, individual scales are not visible over the tibia itself, although darkly pigmented, anteroposteriorly-oriented stripes are densely distributed on the surface of the bone, matching the stripped pattern seen on other parts of the lower leg[19] (Fig. 5a).

On the posterior and anterior surfaces of the ankle, including the dorsal surface of the tarsus, the scales are larger than most other parts of the body ($\bar{x}$ diameter = 3.1 mm) and are distinctively diamond shaped. The relatively large size of these scales and the somewhat "swollen" appearance of the integument (particularly posterior to the ankle joint) invokes the presence of a distinct callosity around the ankle (Fig. 5b). The dorsal surface of the pes and lateral surfaces of the leg are not clearly exposed on either hindlimb. The plantar surfaces of the tarsus and pes are covered in ovoid reticulate scales ($\bar{x}$ diameter = 1.2 mm). This pattern is superimposed on the plantar surface of the metatarsals themselves as well as the soft tissue adjacent to the bones (Fig. 5b). An arthral digital pad arrangement—in which the interpad creases do not correspond to the interphalangeal joints —appears to be present on the fourth pedal digit, which is the only digit where the toe pads are observable (Fig. 5c). Soft tissues are not preserved on the distalmost phalanges, nor can the keratinous ungual sheaths be seen, either because of damage/preparation or due to the position of the animal.

**Tail and cloaca**. The caudal vertebrae are centrally positioned along the long axis of the fleshy part of the tail. The soft tissues maintain a dorsoventral height equal to ~5.5 times the height of the corresponding centrum, increasing to ~6.3 times around caudal vertebra 19, posterior to which the soft tissues and vertebrae are incompletely preserved/truncated. The distal ends of the ischia are covered by comparatively large, polygonal scales that range from subtriangular, rounded-rectangular to hexagonal and forming an ischial callosity ~4 cm in diameter[19]. Scales are

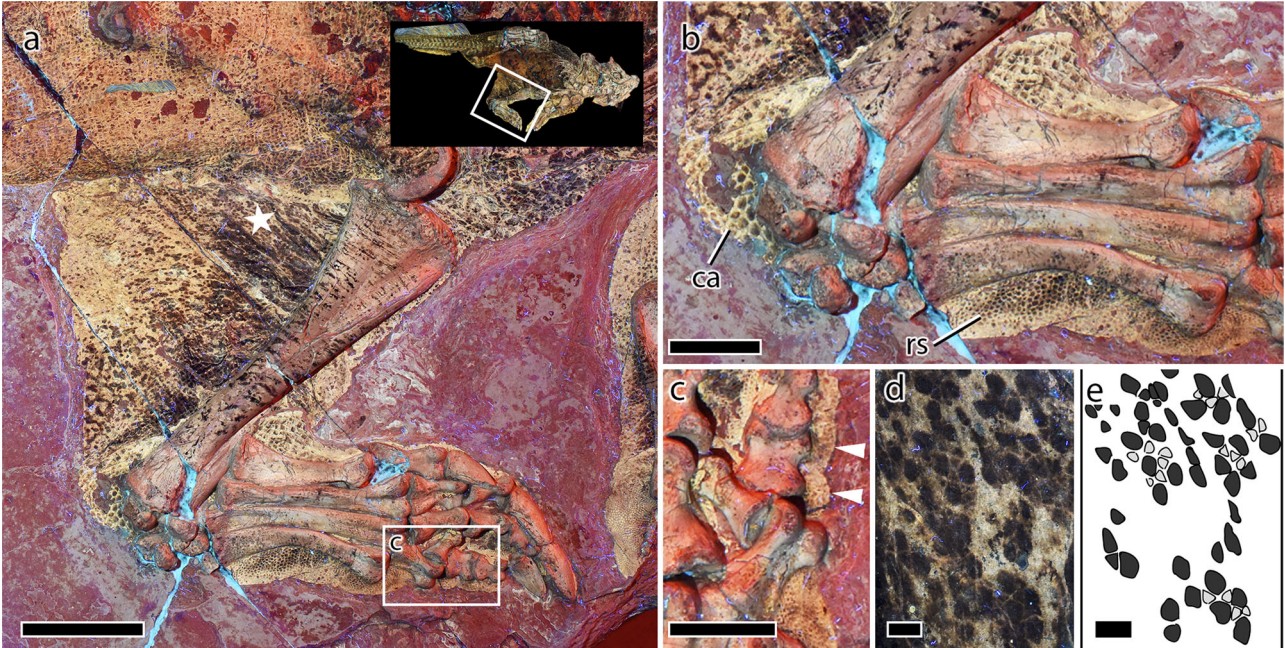

**Fig. 5 Hindlimb and integument of *Psittacosaurus* SMF R 4970 under LSF. a** Left hindlimb in medial aspect with inset showing the region depicted in **a**. **b** Close up of the left pes showing large scales forming a callosity around the ankle joint and reticulate scales covering the plantar surface of the metatarsus. **c** Detail of the fourth pedal digit (boxed region in **a**) showing undulations in the integument corresponding to arthrally arranged plantar pads (arrowheads). **d**, **e** Detail of the scales on the inner leg (star in **a**) showing hexagram pattern of light-colored triangular scales and darker central scales. ca callosity, res reticulates scales. Scale bars equal 5 cm (**a**), 2 cm (**b**, **c**), and 2 mm (**d**, **e**).

largest (up to 6.8 mm diameter) at the center of the callosity, decreasing in size ($\bar{x}$ diameter = 3.2 mm) centrifugally (Fig. 6). A small number of scale inclusions are present on the callosity. Immediately posterior to the ischial callosity is the fleshy aperture (or vent) of the cloaca (Fig. 6). Vinther et al.[25] described the cloaca as consisting of slightly protruding left and right lateral lips that converge anteriorly, forming an inverted "V". Importantly, LSF resolves the anterior convergence of these lips as continuing anteriorly in a straight line for a length of ~2 cm, which we interpret as forming a longitudinally-oriented vent. Thus, the cloaca is shaped more like an inverted "Y". The lateral lips are darkly pigmented and wrinkled, the creases of which are roughly parallel and extend in a posterolateral direction (~3 cm long) towards the ventral tip of the second hemal arch (Fig. 6). Rock breakage above this point (i.e., between the hemal arches themselves) has obscured further details of the integument in this region. Posterior to the lateral lips is another protruding region identified as the dorsal lobe[25]. The dorsal lobe is pale in color (less pigmented) but also wrinkled; wrinkles are parallel, posteriorly oriented and 1–3 cm in length. Scales covering the lateral lips and dorsal lobe are almost lenticular ($\bar{x}$ length = 3.4 mm), the long axes of which are oriented parallel to the surrounding wrinkles, thereby forming a radial pattern around the vent (Fig. 6). Posterior to the cloaca, and for the remaining length of the tail, the integument ventral to the caudal vertebrae consists of vertical bands (mediolaterally oriented in life) of typically rounded-quadrangular scales (Fig. 7). The shapes and sizes of these scales are, however, variable along the length of the tail. Between caudal vertebrae 10–12, the scales increase in size towards the ventral margin of the tail (from $\bar{x}$ = 1.7 to 3.3 mm in height; Fig. 7d). Further distally (between caudal vertebrae 14–15), scales are more uniformly large and rounded-quadrangular ($\bar{x}$ height = 2.8 mm). On the preserved distal part of the tail (i.e., immediately ventral to the hemal arches between caudal vertebrae 16 and 21), the vertical banding of scales ($\bar{x}$ = 2.4 mm in height) is broken dorsally by a longitudinal row of much larger quadrangular feature scales

(Fig. 7b, c). None of these feature scales are complete although they have an average anteroposterior length of 5.4 mm and a maximum height exceeding 8 mm, making them among the largest scales on the entire body of SMF R 4970. These form a longitudinal row at a level roughly in line with the ventral edges of the hemal arches, but whether or not they continued more dorsally or constituted any more than a single row is unknown as the scales are not well preserved over the surface of the bones. Dorsal to the caudal vertebrae, individual scales are difficult to discern, obscured by dark patches of pigmentation[19]. Although pigment spots approximate the size and, to a lesser extent, shape of epidermal scales on other parts of the tail, there is not necessarily a direct relationship between pigmentation and scale morphology[19]. Indeed, on the dorsal part of the proximal tail (between caudal vertebrae 1–5) where pigmentation is dense (80% coverage[19]) and scales are better preserved, pigmented regions span 10 or more adjacent scales, separated by unpigmented strips one or two scales wide. Scales in both the pigmented and unpigmented regions appear subcircular/ovoid ($\bar{x}$ = 2.8 mm in diameter). Smaller inclusions are also present but there is no evidence of the vertical banding or larger feature scales seen on the ventral and distal parts of the tail.

The bristles lining the dorsal margin of the tail between caudal vertebrae 5–19 were described in detail by Mayr et al.[23] (Fig. 7a, b). It is not necessary to repeat those descriptions here.

SMF R 4970 is remarkable for the extent of soft tissue preservation, which has revealed unprecedented integumentary structures including the dorsal row of bristles on the tail, the cloaca, umbilical scar, as well as evidence of countershading[16,19,23–25]. Additional integumentary features either not previously recognized or expounded upon include: (1) the keratinous jugal "horn"[23]; (2) enlarged scales of the ischial callosity[19,25]; (3) the row(s) of feature scales on the mid-distal tail, and; (4) the arthral arrangement of the digital pads. We elaborate on each of these structures and discuss additional findings on the cloaca and the general skin morphology in ceratopsians below.

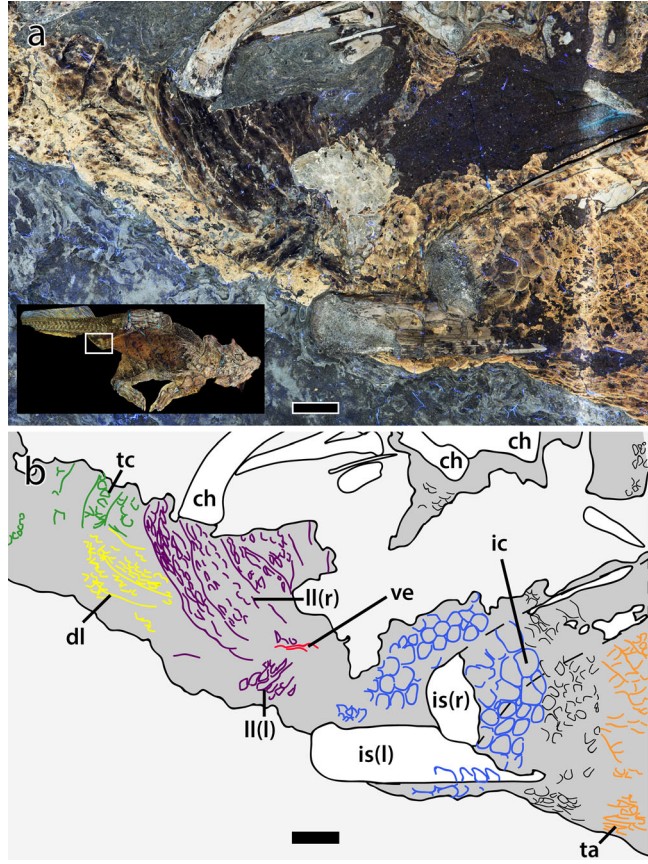

**Fig. 6 Cloaca of *Psittacosaurus* SMF R 4970 under LSF. a** Photograph and **b** interpretive illustration with inset showing the region depicted in **a** and **b**. Colors in **b** depict skeletal elements (white); integument (dark gray); lateral lips (purple) and dorsal lobe (yellow) of the cloaca; cloacal vent (red); ischial callosity (blue); transversely-banded caudal scales (green) and; transversely-banded abdominal scales (orange). ch chevron, dl dorsal lobe, ic ischial callosity, is ischium, l left (in brackets), ll lateral lip, r right (in brackets), ta transversely banded abdominal scales, tc transversely banded caudal scales, ve vent. Scale bars equal 1 cm.

**Keratinous jugal horn**. The prominent jugal horn is one of the most characteristic features of *Psittacosaurus* spp., and ceratopsians more generally[30,31]. Anastomosing vascular channels on the surface of the jugal horn in *P. xinjianensis* were cited as probable evidence for a keratinous sheath in life[32]. The taxonomy of *Psittacosaurus* is complex but consensus is starting to emerge[31,33] revealing variation in the presence/absence of such neurovascular channels on the jugal horn[31]. Neurovascular channels are present on both dorsal and ventral surfaces of the jugal horn in *P. xinjiangensis*[32] and *P. gobiensis*[34], present only on the dorsal surface in *P. houi* (= *P. lujiatunensis*[30]), present posteriorly in *P. sibiricus*[35], and entirely absent in *P. meileyingensis*[36]. Intraspecific variation in this feature reported in *P. sibericus* may also be influenced by sex and/or ontogeny[35]. Therefore, taxonomic interpretations based on horn size, form, and orientation in *Psittacosaurus* should be regarded with caution (see ref. [31]). Nevertheless, based on variation in osteological correlates on the jugal[37], the epidermal covering too would be expected to differ between species—and possibly life stages—and consequently not all species of *Psittacosaurus* would have had keratinous sheaths that covered the entirety of the jugal horn. *Psittacosaurus* SMF R 4970 appears to show direct evidence of this: rather than a keratinous "sheath", the right jugal horn preserves evidence of polygonal scales directly on the ventral surface of the bone. On the left side, individual scales are not discernible; however, the dark-colored triangle of soft tissue preserved at a deeper level below the jugal (i.e., anatomically dorsal) almost certainly represents the keratinous "sheath"[23]. Unlike in some previous reconstructions (e.g. ref. [19]), we interpret the ventral surface of the jugal horn in *Psittacosaurus* SMF R 4970 as covered in epidermal scales, whereas the dorsal surface had a keratinous covering, perhaps more analogous to a fingernail than a sheath. This interpretation is also consistent with the osteological correlates of such structures[37]: in SMF R 4970, the relatively smooth, porous bone texture of the ventral jugal horn is not congruent with a thick keratinous covering[37]. Although the dorsal surface is not visible in SMF R 4970, You et al.[30] described the jugal horn of *P. houi* (= *P. major*[23])—a potential candidate for the identity of SMF R 4970 (ref. [19]; Supplementary Information)—as smooth ventrally and bearing vascular grooves dorsally. Thus, based on osteological correlates alone[37], there is a precedent for the condition in which only the dorsal surface of the jugal horn would have had a keratinous covering. This interpretation also explains

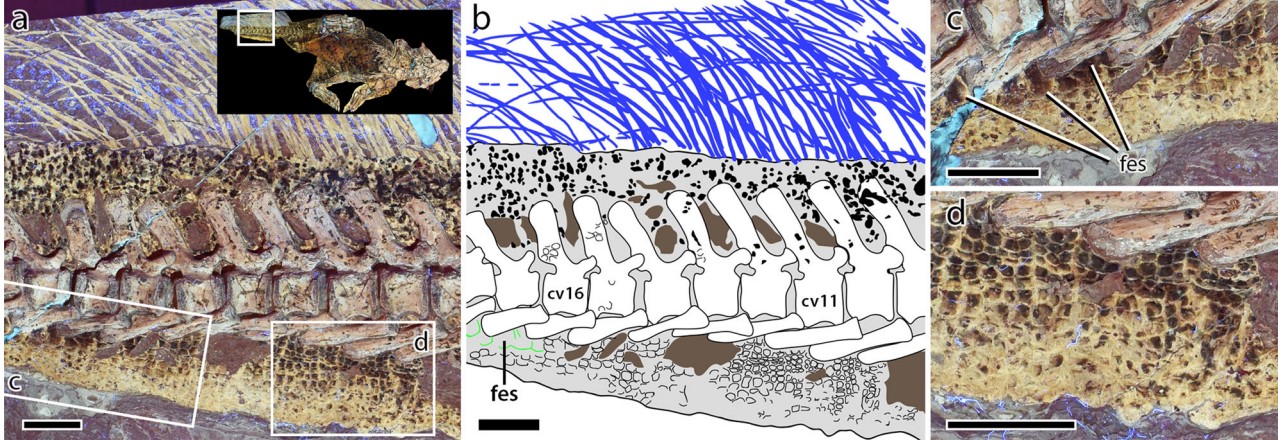

**Fig. 7 Tail of *Psittacosaurus* SMF R 4970 under LSF. a** Photograph and **b** interpretive illustration with inset showing the region depicted in **a** and **b**. In **b**: integumentary outline (gray), tail bristles (blue), matrix (brown), transversely banded epidermal scales (thin black outlines), large quadrangular feature scales (green outline), dorsal pigmentation (black). Close up of the feature scales (**c**) and transversely banded epidermal scales (**d**). cv11 11th caudal vertebra, cv16 16th caudal vertebra, fes feature scales. Scale bars equal 2 cm.

the preservation of the keratinous jugal "horn" in SMF R 4970 in which it appears to have shifted anteriorly (and perhaps laterally) from the jugal horn[23]. Such a shift would be difficult to reconcile had the jugal horn been entirely sheathed in keratin. As also acknowledged by those authors, it is possible that the dark triangle was not associated with the jugal at all, but from a more dorsal position on the skull[23], which we agree with. We nevertheless interpret the presence of a unique keratinous plate or nail-like covering in SMF R 4970 and likely *P. houi*[30], and which probably differed in other species of *Psittacosaurus*—and potentially at different life stages—based on differing bone textures on the jugal horn.

Although the jugal horn and the keratinous "sheath" are misaligned[23], preservation of the latter in *Psittacosaurus* permits some tentative estimations on horn dimensions in other ceratopsians. By following the preserved curvature of the margin of the bony core and its horn "sheath", we estimate that the latter is around 140% larger than the bony core in SMF R 4970. This value is greater than in the largest osteoderms in ankylosaurian *Borealopelta* (125% for the parascapular spine[38]), but within the range of some modern bovids such as the bison and bull[38]. Applying the same value (140%) to other ceratopsian horn cores suggests some of the largest postorbital horns (e.g., *Triceratops*, estimated horn core length = 115 cm in MOR 3027 using Scannella et al.[39], Fig. S1) might have been over 1.5 m in length. Such extrapolations should, however, be seen as highly tentative given that the "sheath" in SMF R 4970 has evidently shifted and that the relative proportions of the jugal horn of *Psittacosaurus* and that of nasal and postorbital horns of ceratopsids might be significantly different. Keratinous contributions to horn length are also highly variable in mammals such as bovids (see Supplementary Data), therefore we consider it unreliable to use the length of the jugal horn of a single *Psittacosaurus* specimen to infer horn lengths in other ceratopsians. Nevertheless, these ranges illustrate the importance of soft tissues in enhancing the external appearance of even comparatively modest horns, such as the jugal horn in *Psittacosauru*s.

**Ischial callosity**. The enlarged scales forming the ischial callosity have been remarked upon previously[19,25]. Other ornithischian examples of an ischial callosity are unknown but are occasionally preserved in the theropod ichnological record. In rare ichnites of "sitting" theropods (*Kayentapus*, *Grallator* and *Eubrontes*), the ischial callosity may impress as a circular, crescentic or subtriangular depression[40–42]. In one specimen of *Grallator* that preserves skin impressions from the Lower Jurassic Turners Falls Formation, Kundrát[41] (p. 359) reported the ischial callosity bore "small trapezoid scales…similar in size to those preserved on the pedal surface of *Anomoepus intermedius*". From this description, the ischial callosity in *Grallator* differs from the "reinforced" condition in *Psittacosaurus*. Enlarged scales in this region of *Psittacosaurus* (Fig. 6) would have reduced the surface area of interstitial skin (i.e., hinge areas) exposed and therefore helped protect these soft parts from abrasion. Melanisation of the scales covering the ischial callosity might also have played a role in structural strengthening of this region[19] but could also have played a visual role. As in bipedal theropods, it can be reasonably concluded that the ischial callosity in *Psittacosaurus* would have been used to support part of the animal's weight when sitting or crouching[19].

**Caudal feature scales**. At least one longitudinal row of feature scales is present on the lateral part of the tail between caudal vertebrae 16 and 21 (Fig. 7a–c). These do not have the same raised morphology of the feature scales in the pectoral region but are plate-like and more similar to (albeit much larger than) the ventral scales on the tail. We do not consider these scutate ventral scales (e.g., as seen in the tail of the theropods *Concavenator* and *Juravenator* or lining the ventrum of living snakes[43–45]) as they do not occur along the ventral midline, which can be traced posteriorly based on the position of the cloaca and ischial callus, and which indicate a lateral, or ventrolateral, position on the tail for the feature scales (ref. [19] Supplementary Information). Similar rows of enlarged scales have also been described in the embryonic Auca Mahuevo titanosaurs[46,47] and along the tail of the early-branching ornithischian *Kulindadromeus*[48,49]. Colouration in SMF R 4970 may also belie the function(s) of these scales, although their ultimate role remains equivocal. Relatively large scales reduce the exposure of softer interstitial skin and afford greater protection compared to small scales[45,50]. However, the uniformly dark colouration of the feature scales—in comparison to the pale ventral and more mottled dorsal parts of the tail[19]—might also imply a visual function. Dark feature scales occur in close association with the elaborate row of 'bristles' on the dorsal tail, which has been interpreted as a display device for sociosexual signaling[16,23]. Dark feature scales would have added additional visual contrast to the tail and, combined with striped feature scales on the chest and forelimb and an overall countershaded body colouration[19], invoke an unusual and visually striking animal.

**Digital pads**. *Psittacosaurus* presents an arthral arrangement of the pads on the pedal digits (those of the manus are not preserved; Fig. 5c). Several authors have proposed that the arthral condition—in which the interpad crease does not align with the interphalangeal joint—is the primitive condition among dinosaurs[51], which is supported by its presence in a number of non-avian theropods and basal birds[52,53]. More crownward birds evolved a mesarthral condition, in which the interpad crease corresponds to the interphalangeal joint; however, both conditions are present in extant birds and can vary among individuals[54]. In non-avian theropods, an arthral arrangement is present in allosauroids (*Concavenator*[55]), tyrannosauroids (*Santanaraptor*[6]), and maniraptorans (*Sinornithosaurus* [GMC 91, STM 5-172]; *Anchiornis* [STM-0-7][6,29]). To our knowledge, *Psittacosaurus* is the first ornithischian to preserve direct evidence of the arthral pad configuration. This configuration is not directly observable from footprints and the few body fossils with skin in this region (e.g., *Corythosaurus*, *Kulindadromeus*) do not reveal the shapes of the pads[49,56]. Arthrally-arranged pads do, however, appear to be present on the manus of the early-branching ornithischian *Kulindadromeus* (ref. [49]; fig. 4.5a, c). The identification of arthral pads in *Psittacosaurus*, therefore, upholds Rainforth's[51] hypothesis that all dinosaurs retained the arthral condition in the pes, at least plesiomorphically.

**Cloaca**. One of the most surprising features of SMF R 4970 is the preservation of the cloaca, which was recently described by Vinther et al.[25]. Those authors identified the unique V-shaped convergence of the lateral lips and the presence of a bulbous dorsal lobe but were unable to decipher the precise shape of the opening (or vent) under white light. LSF resolves this issue and clearly shows the vent as a longitudinal slit, ~2 cm long, anterior to the dorsal lobe, between the left and right lateral lips (Figs. 6 and 8c). This is notable, as the shape of the vent in living sauropsids has taxonomic relevance and is accompanied by various configurations of the internal anatomy of the cloaca (see Supplementary Data). The cloaca of living sauropsids is divisible into three types[57]: transversely opening (snakes and lizards), longitudinally opening (crocodylians; Fig. 8b, d), or round/square

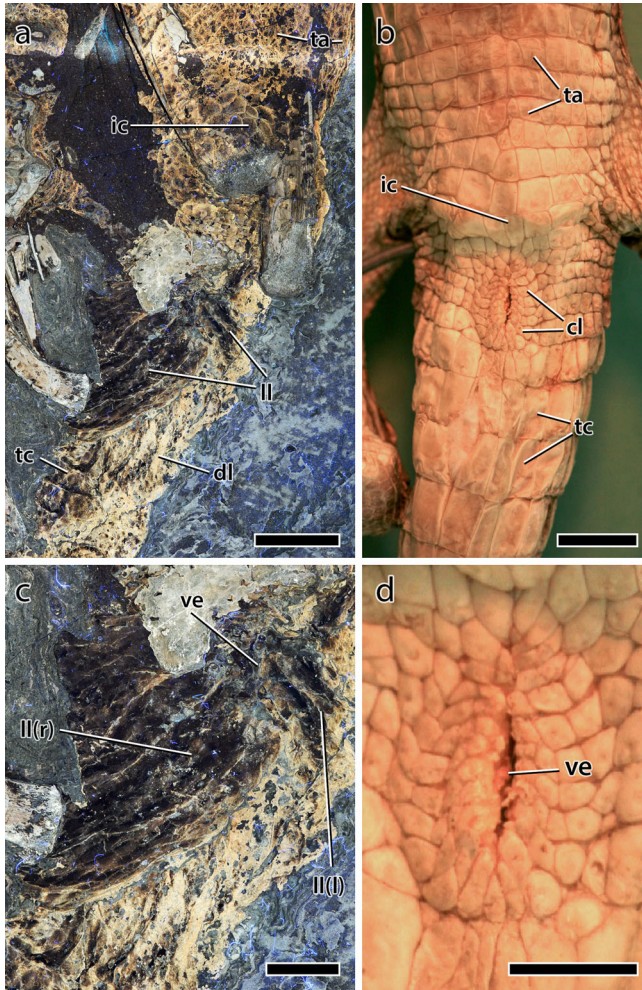

**Fig. 8 Cloaca of *Psittacosaurus* and *Crocodylus*. a** Cloacal region of *Psittacosaurus* (SMF R 4970) under LSF showing the ischial callosity, transversally banded abdominal and caudal basement scales, and lateral lips of the cloaca. **b** Cloacal region of a juvenile freshwater crocodile (*Crocodylus johnsoni*; University of New England Natural History Museum, no specimen number) showing the rosette pattern of scales surrounding the vent and transverse banding of the ventral scales on the abdomen and tail. **c** Close up on the cloaca of *Psittacosaurus* (SMF R 4970) under LSF showing the two lateral lips and the longitudinally-oriented cloacal vent. **d** Close up on the cloaca and longitudinally-oriented vent of *Crocodylus johnsoni*. cl cloacal scales, dl dorsal lobe, ic ischial callosity, l left (in brackets), ll lateral lip, r right (in brackets), ta transverse abdominal scales, tc transverse caudal scales, ve vent. Anterior is toward the top in all images. Scale bars equal 2 cm (**a**), 1 cm (**b**, **c**), and 5 mm (**d**).

(birds; see Supplementary Data). Our observations reveal that the integumentary covering across these three types differs accordingly. In snakes, the transverse vent is covered by one or two cloacal scales that are modifications of the broad ventral scales present elsewhere on the underbelly (see Supplementary Data). The scalation pattern in lizards is highly variable but the vent is always transverse and accompanied by a variable number of cloacal scales that may or may not differ from the surrounding scales. Among birds, the area immediately surrounding the cloaca is naked, bearing neither scales nor feathers. In crocodylians, the longitudinal vent is surrounded by elliptical-to-polygonal scales that radiate and increase in size from the vent itself (Fig. 8d). This rosette arrangement of cloacal scales was not observed in any squamate and is distinct from the transverse rows of

comparatively large quadrangular scales in crocodylians that extend along the ventral surfaces of the abdomen and tail (Fig. 8b). Despite the difference in the configuration of the lateral lips and dorsal lobe[25], the gross morphology of the vent and surrounding scales in *Psittacosaurus*—which combines a longitudinally opening vent with a rosette pattern of cloacal scales and transverse rows of quadrangular ventral scales on the ventral tail and abdomen (Figs. 6 and 8a, c)—most closely matches that of crocodylians (Fig. 8b, d).

The internal anatomy of the cloaca also differs between crocodylians, squamates, and birds, which correspond to the three cloacal morphotypes (longitudinal, transverse, round/square, respectively[57–60]; see Supplementary Data). Therefore, the longitudinal vent of *Psittacosaurus* potentially implies a crocodylian-like internal anatomy of the cloaca. In archosaurian and lepidosaurian reptiles, the cloaca forms the common opening of the digestive and urogenital tract and consists of a series of chambers—the coprodeum, urodeum, and proctodeum—separated by muscular sphincters and which terminates in the vent[58,60]. The coprodeum, the most proximal of the chambers, receives waste from the intestines, the urodeum receives products from both the genital and urinary ducts, and the proctodeum houses the male copulatory organ. Squamates follow this general pattern although the copulatory organ—the paired hemipenes—is unique and dorsally situated within the proctodeum[57,58]. In contrast, crocodylians, and some birds possess a single, ventrally-positioned copulatory organ, but the majority of birds lack a phallus entirely[59,60]. Also in contrast to squamates and late-diverging birds (Neognathae), the ureter in crocodylians opens into the coprodeum, rather than the urodeum[60]; a condition also found in some palaeognaths (e.g., *Rhea*, tinamous[60]). Based solely on the external anatomy of the vent in *Psittacosaurus* and its similarity to crocodylians, we hypothesize the presence in the former of a muscular, unpaired, and ventrally-positioned copulatory organ (e.g. ref. [61]) and a ureter that empties into the coprodeum[60], which is consistent with prior studies based on the extant phylogenetic bracket[62,63]. Like crocodylians, birds also use internal fertilization (regardless of the presence of a phallus), which is the presumed method in *Psittacosaurus*[63], although the sex of SMF R 4970 cannot be determined at present[25]. The presumably paired oviducts in *Psittacosaurus*[64] would have opened into the urodeum as well.

**Skin morphology in ceratopsian dinosaurs.** Ceratopsia is a taxonomically extremely diverse group (>60 genus-level taxa) of herbivorous ornithischians from the Middle Jurassic to the Late Cretaceous characterized by a toothless and keratin-covered beak and, in most members, a frill extending over the rear of the skull and horns at the level of the cheek, nose and/or eyes[65–68]. Despite this diversity, the preserved integument is currently known from a handful of specimens (<20) and restricted to six ceratopsian genera (Supplementary Table 2 and Fig. 9; see Supplementary Data). *Psittacosaurus* is the only non-coronosaurian ceratopsian with preserved integument, which has been described or reported in six specimens (including three exquisitely preserved and nearly complete specimens)—possibly seven if GMC LL2001-01 (Fig. 10) represents the psittacosaurid *Psittacosaurus*—, making it the ceratopsian with the most extensively preserved integument[16–19,21,27] (Fig. 9 and Supplementary Table 2; see Supplementary Data). Within Coronosauria, squamous skin is ostensibly preserved in the protoceratopsid *Protoceratops*[15], and is definitively present in the centrosaurines *Centrosaurus* (AMNH FARB 5351, AMNH FARB 5427, TMP 1986.018.0097[11,12]) and *Nasutoceratops* (UMNH VP 16800[14,22]), and in the chasmosaurines *Chasmosaurus* (CMN 2245, UALVP 52613, FHSM VP-117[12,13,69,70]) and *Triceratops* (HMNS PV.1506; CMN

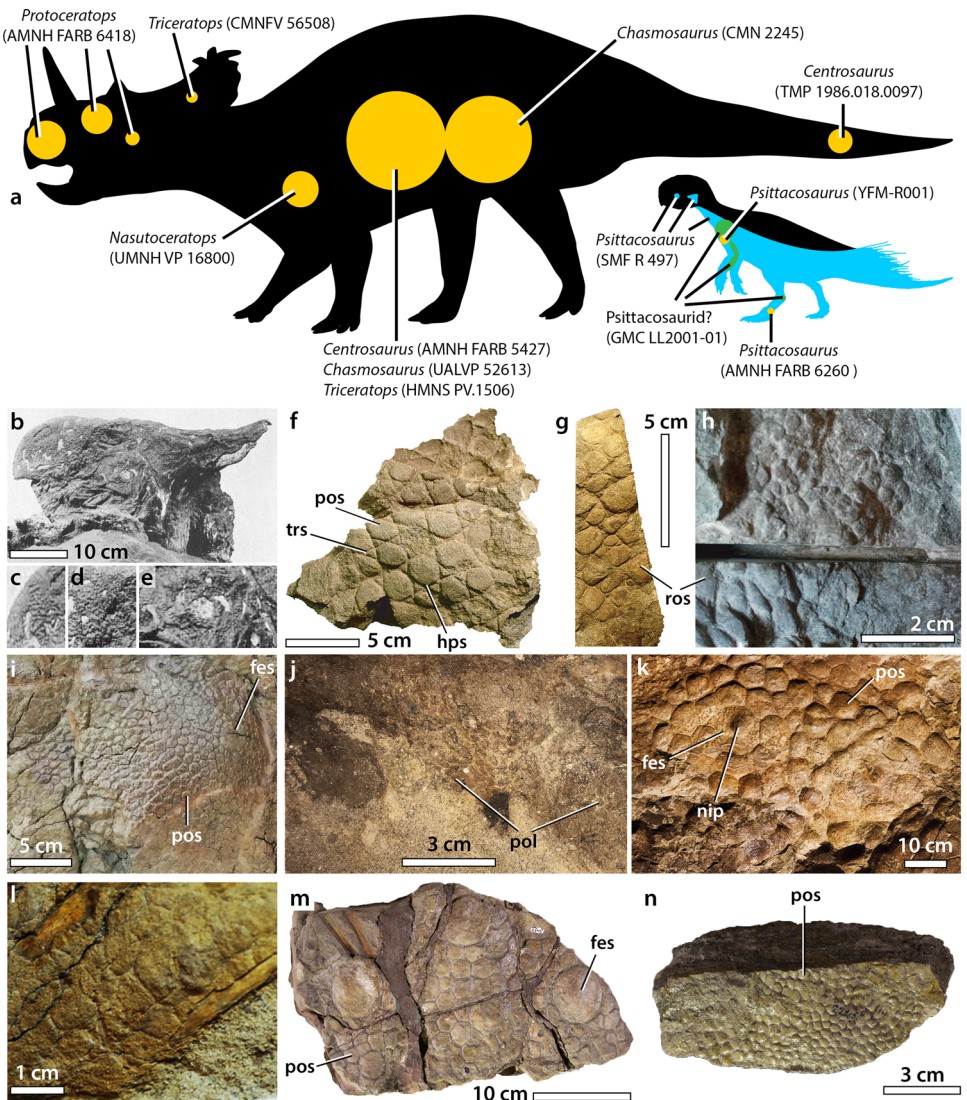

**Fig. 9 Integumentary structures in ceratopsian dinosaurs. a** Distribution of known scaly integument on the body of psittacosaurids (right silhouette; Jaime Headden; https://creativecommons.org/licenses/by/3.0/; modified) and coronosaurian ceratopsians (left silhouette; Caleb M. Brown; https://creativecommons.org/licenses/by-sa/3.0/; modified), with the position of the integument in the *Psittacosaurus* specimen SMF R 497, the putative psittacosaurid specimen GMC LL2001-01, and the other ceratopsian specimens in blue, green and orange, respectively. Mummified head covered with skin possibly made of minute pebbly basement scales in the protoceratopsid *Protoceratops* (AMNH FARB 6418; from Brown and Schlaikjer[15], modified) in left lateral view (**b**), with close up on putative basement scales of the beak (**c**), the lacrimal part (**d**), and the cheek (**e**). Patches of skin made of triangular, polygonal or subcircular basement scales from the proximal forearm (**f, h**) and shoulder region (**g**) of the left forelimb of the centrosaurine *Nasutoceratops titusi* (UMNH VP 16800; courtesy of Erik K. Lund). **i** Patches of skin made of feature and polygonal basement scales from the thoracic region of the centrosaurine *Centrosaurus apertus* (AMNH FARB 5427; courtesy of Carl Mehling). **j** Polygonal basement scales from the distal tail region of the centrosaurine *Centrosaurus* sp. (TMP 1986.018.0097; courtesy of Caleb Brown). **k** Polygonal feature and basement scales from the flank of the chasmosaurine *Triceratops horridus* (HMNS PV.1506; courtesy of Marschal A. Fazio). **l** Polygonal basement scales in a juvenile individual of the chasmosaurine *Chasmosaurus belli* (UALVP 52613; courtesy of Philip J. Currie). Two patches of skin, one with large rounded feature scales surrounded by smaller polygonal basement scales (**m**) and a second with small polygonal basement scales (**n**), from the pelvis arch and right flank of an adult specimen of the chasmosaurine *Chasmosaurus belli* (CMN 2245; from S. E. Pan/Canadian Museum of Nature, used under CC BY-NC 4.0; These images were cropped from the original). fes feature scale, hps hexagram pattern of basement scales, nip nipple-like structure on the feature scale, pos polygonal basement scale, ros rounded basement scale, trs triangular basement scale.

FV 56508[20]; Supplementary Table 2; see Supplementary Data). *Chasmosaurus* is the only ceratopsid with skin from both juvenile and adult individuals[13,70].

The skin of ceratopsians is best represented on the flank, hindlimb and pelvic regions (Fig. 9), being preserved in these parts of the body in *Psittacosaurus* (SMF R 4970), *Centrosaurus* (AMNH FARB 5351 [holotype of *Monoclonius nasicornis*]; AMNH FARB 5427 [holotype of *Monoclonius cutleri*]), *Chasmosaurus* (CMN 2245; UALVP 52613), and *Triceratops* (HMNS

PV.1506). *Nasutoceratops* (UMNH VP 16800) preserves skin on the brachium and shoulder regions, whereas most of the integument is known for *Psittacosaurus*, including multiple specimens that preserve skin on the flank (MV 53; SMF R 4970), shoulder (SMF R 497; YFM-R001), pedes (AMNH FARB 6260; SMF R 4970) and tail (PKUP V1051; SMF R 4970). Besides *Psittacosaurus* (SMF R 4970; Fig. 2a–d), possible skin covering the head has only been found in *Protoceratops* (ref. [15] plate 13) and *Triceratops* (CMN FV 56508; J. Mallon, pers. comm. 2021),

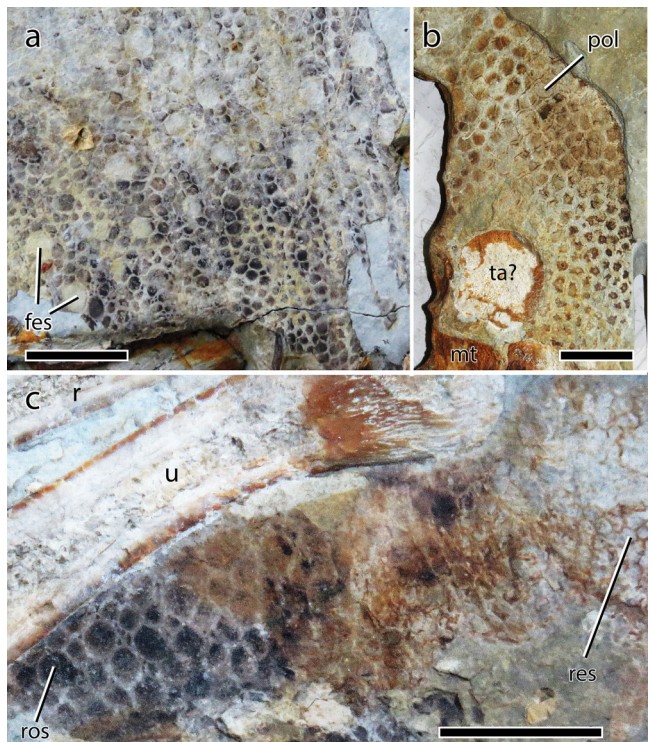

**Fig. 10 Integument in a possible psittacosaurid specimen (GMC LL2001-01) from the Dawangzhangzi Village of Western Liaoning, China.**
**a** Rounded feature scales and polygonal basement scales from the shoulder and possibly neck regions. **b** Polygonal basement scales proximal to the metatarsals. **c** Polygonal basement and reticulate scales posterior to the distal half of the right ulna. fes feature scale, mt metatarsus, pol polygonal basement scale, r radius, res reticulate scale, ros rounded basement scale, ta? possible tarsus, u ulna. Scale bars equal 10 mm.

although these have yet to be formally described. Additional specimens preserving squamous skin in *Psittacosaurus*[26], *Centrosaurus* (AMNH FARB 5351), a juvenile *Triceratops*[71] and a particularly complete specimen of *Triceratops horridus*[20] have also never been illustrated nor described in detail. Three large sections of skin belonging to the *Triceratops* specimen (HMNS PV.1506) reported by Larson et al.[20] are, however, on display at the Houston Museum of Natural Science; our observations on *Triceratops* integument rely on these specimens, pending a thorough description of the material by Larson et al.

Like many dinosaurs[4,6,10,72], ceratopsian skin typically consists of subcircular-to-polygonal feature scales surrounded by a network of low and smaller non-overlapping polygonal basement scales separated by narrow interstitial tissue; however, the basement scales are usually relatively larger than in ornithopods and theropods (Fig. 11; see Supplementary Data). Important variations in scale size, shape and pattern also occur between ceratopsian taxa and over different body parts. *Psittacosaurus* (SMF R 4970) is to our knowledge the only non-ceratopsid ceratopsian with a preserved keratinous horn "sheath", which is ~140% larger than the bony core of the jugal horn (see above). Several authors have, however, reported the presence of a horn "sheath" in other ceratopsids. American paleontologist John Bell Hatcher was the first to report such a discovery in the *Triceratops* specimen YPM 1821, writing that "a portion of the investing horny material was still in place about the left horn core, though in such a decomposed condition that it was impossible to preserve it." (Hatcher et al.[73], p. 32). Likewise, Czerkas[2] briefly mentioned the probable remains of the outer sheath—consisting

of a carbonaceous powdery layer up to two centimetres thick—in a young *Triceratops* skull. More recently, Happ[74] reported the discovery of a claystone layer grading from 7 to 33 mm thick and distinct in composition from the bony core in the left postorbital of an adult specimen of *Triceratops* (SUP 9713.0). This mineralized layer, which covers a 1.2–5.3 mm thick outer bone layer composed of compact Haversian bone, is interpreted by Happ[74] as a replacement of the horn sheath.

Other than *Psittacosaurus*, skin from the head has not been formally described for any ceratopsian although there are several reports. In *Protoceratops* (AMNH FARB 6418), a thin, wrinkled layer of matrix covering a large portion of the cranium and mandible was interpreted as skin by Brown and Schlaikjer[15] (Fig. 9b–e). Presumably based on the photos published by Brown and Schlaikjer[15] (ref. [15] plate 13; Fig. 9b–e), Czerkas[2] also considered the presence of desiccated and sunken eyelids; however, the presumed integument has since been entirely prepared off the specimen and verification of any of these interpretations is no longer possible. Davis[75] also reported skin covering the head of a possible *Triceratops* comprising large circular feature scales surrounded by smaller polygonal basement scales based on a photograph in Lessem[76] (p. 41). However, the photograph is not from the skull of *Triceratops* (Sylvia Czerkas, pers. comm. May 2021) but from the flank of *Chasmosaurus* (CMN 2245; C. H. pers. obs.). A small piece of skin associated with the frill of the *Triceratops* specimen CMN FV 56508 (J. Mallon, pers. comm. 2021) in fact reveals that the frill of this taxon, and probably all ceratopsians, was covered with small polygonal basement scales, rejecting Horner and Marshall's[77] hypothesis that a keratinous sheath covered nearly the entire skull of ceratopsians such as *Triceratops* and *Torosaurus*. Using osteological and histological correlates in extant amniotes, Hieronymus et al.[37] showed that several rows of epidermal scales were present on the surface of the cranium in centrosaurine ceratopsids, namely, a median row of shallow scales on the parietal bar (*Centrosaurus*, *Achelousaurus*, *Pachyrhinosaurus*), a series of scales lining the dorsal rim of the orbit and onto the squamosal (*Centrosaurus*, *Einiosaurus*), a second row of scales anteroventral to the former on the squamosal (*Centrosaurus*), and a midline row of epidermal scales between the horny beak and the nasal boss (*Pachyrhinosaurus*).

Scaly integument from the neck is known from *Psittacosaurus* (SMF R 4970; Fig. 2g–h) and possibly from the putative psittacosaurid specimen GMC LL2001-01, in which a ~36 cm² patch of skin comes from the neck and/or the shoulder regions[21]. The skin of GMC LL2001-01 consists of numerous small (2.5–3 mm) subcircular feature scales surrounded by a mosaic of minute (1–1.5 mm) basement scales (Fig. 10a), a pattern similar to that seen on the shoulder of SMF R 4970, suggesting that it likely comes from the same body region. The presence of numerous feature scales on such a large patch of skin, however, indicates that they were not restricted to the pectoral region but probably covered part of the neck dorsal to the shoulder. The feature scales are separated from each other of a distance of 3 to 5 mm[21] and although their size and distribution on the skin appear to be random, two parallel rows of pseudo-aligned feature scales are present in one corner of the patch (Fig. 10a). Notably, the feature scales lack the striped pigmentation seen in *Psittacosaurus* (SMF R 4970), but given the taxonomic ambiguity of GMC LL2001-01, we cannot comment on the relevance of this discrepancy. The basement scales are rounded and irregular or vaguely polygonal in shape (tetragonal, pentagonal, or hexagonal according to ref. [21]). Relatively large basement scales, usually seven to ten, encircle the feature scales. A single or a group of two-to-four small (<1 mm) irregular, subcircular or triangular basement scales, probably representing scale inclusions, are present between the larger basement scales in a few areas of the patch (Fig. 10a).

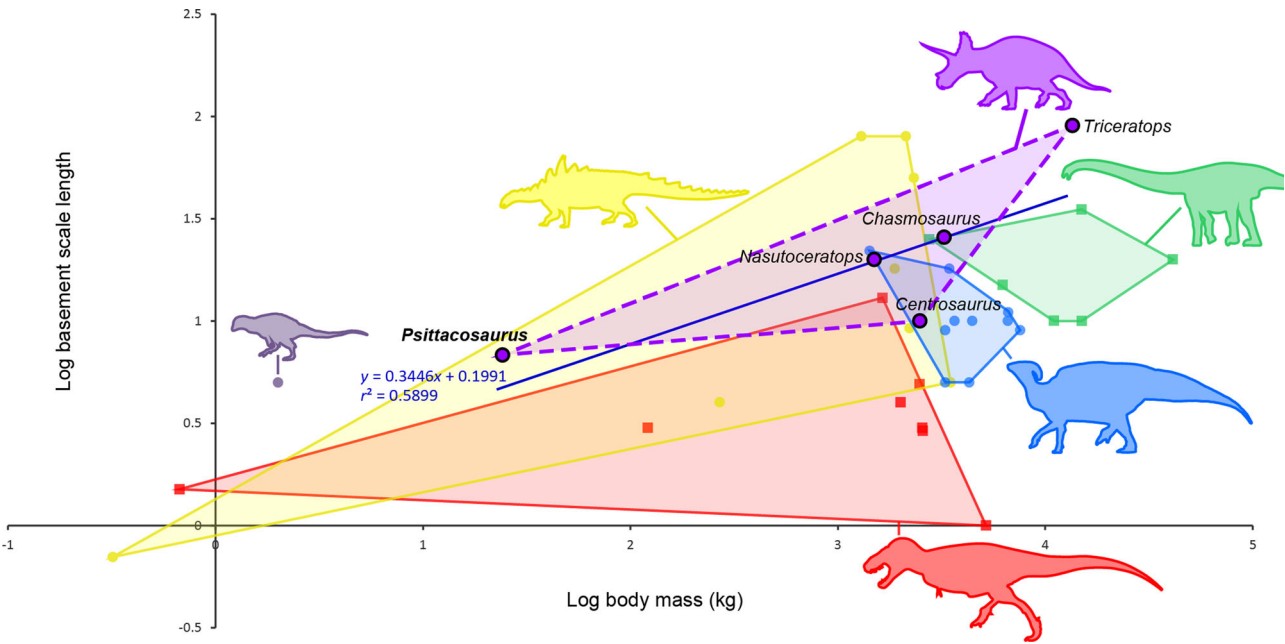

**Fig. 11 Basement scale length vs. body mass in non-avian dinosaurs.** Log-plot of basement scale length (taken from the largest scale) versus body mass in ceratopsians (purple dots; silhouette: Scott Hartman; https://creativecommons.org/licenses/by-nc-sa/3.0/; modified), *Kulindadromeus* (violet dot; silhouette: Pete Buchholz; https://creativecommons.org/licenses/by-sa/3.0/; modified), ornithopods (blue dots; silhouette: Matt Martyniuk; https://creativecommons.org/licenses/by-nc-sa/3.0/; modified), thyreophorans (yellow dots; silhouette: Scott Hartman; https://creativecommons.org/licenses/by/3.0/; modified), non-avian theropods (red squares; silhouette: Scott Hartman; https://creativecommons.org/licenses/by-nc-sa/3.0/; modified), and sauropods (green squares; silhouette: Scott Hartman; https://creativecommons.org/licenses/by-nc-sa/3.0/; modified). Ceratopsian taxa are labeled for clarity. The regression equation and $r^2$ value are reported for ceratopsians. Figure modified from Hendrickx and Bell[78] (ref. [78]: Fig. 13f). Data on body masses and basement scale length is provided in the Supplementary Data.

In the *Psittacosaurus* specimen YFM-R001, the pectoral girdle and forelimb is covered with small (3–5 mm), polygonal (4–6 sided) basement scales and minute (1–2 mm) triangular scales[27], although there is no indication of the raised feature scales seen in this area of SMF R 497. Whether this difference is due to intra- or interspecific variation or some other factor is unknown. The hexagonal and triangular scales in YFM-R001 together form a hexagram pattern anterior to the mid-shaft of the humerus (ref. [27]; Fig. 2), identical to that on the posterior part of the brachium and hindlimb of SMF R 497 (Fig. 3c). This pattern is also seen in a patch of skin in GMC LL2001-01 directly posterior to the humerus, at one third of the bone's length. It is also present on the brachium of *Nasutoceratops*, where relatively large (8–12 mm) hexagonal basement scales are framed by six small (3–4 mm) triangular scales (patch C of refs. [14,22]; Fig. 9f). As scales with a hexagram arrangement occur in *Psittacosaurus* (SMF R 497, brachium, inner thigh; YFM-R001, brachium), the possible psittacosaurid specimen GMC LL2001-01 and *Nasutoceratops* (brachium), such a pattern might have been common on the limbs of ceratopsians. It is worth noting that the hexagram pattern differs from the multi-pointed feature scales seen in some hadrosaurids[9] and that the former appears to be an integumentary design unique to ceratopsians. Other types of scales from the shoulder region include proximodistally elongate polygonal or rounded-polygonal (3–6 sided) scales in SMF R 497 (Fig. 3b), and, in *Nasutoceratops*, medium to large (10–20 mm) subcircular, elliptical or rhomboid basement scales arranged in irregular rows and surrounded by smaller (5–10 mm) subcircular to triangular scales (patch B of refs. [14,22]; Fig. 9g). *Nasutoceratops* also shows an array of variably-sized (2–8 mm), tightly-packed, oval-to-subcircular scales arranged in irregular rows anterior to the humeral head (patch A of refs. [14,22]; Fig. 9h).

The integument on the rest of the forelimb and manus of ceratopsians is only known in *Psittacosaurus* (SMF R 4970; Fig. 3e) and the putative psittacosaurid specimen GMC LL2001-01 in which several patches of skin are present posterior to the humerus and the ulna[21] (Fig. 10c). Next to the humerus, the basement scales (0.5–1.5 mm) are irregular, subcircular or lenticular in shape, with rounded edges. Unlike in SMF R 4970, a few larger subcircular and ovoid feature scales (1.6–2.2 mm) can be observed in the posteriormost area of the skin, along the proximal two-thirds of the humerus. No feature scales are present at the level of the ulna where the patch of skin consists of relatively large subcircular to lanceolate basement scales (2–3 mm[21]) showing an anteroposterior elongation axis (Fig. 10c). The basement scales diminish slightly in size posteriorly and gradually transform into smaller rounded and polygonal scales distally. A patch of small (~1.5 mm[21]) reticulate scales is clearly visible posterodistal to the distal extremity of the ulna, between this bone and the metacarpals (Fig. 10c). Reticulate scales on the palmar surface of the manus are also found in SMF R 4970 (Fig. 3e); they are not known from ceratopsid body fossils or tracks.

Scales over the flank strongly vary among ceratopsians, but all involve feature scales set within a basement of smaller scales. The feature scales are small (3–4 mm), low, and circular-to-irregular in *Psittacosaurus* (SMF R 4970); larger (50–80 mm), flat or weakly convex, and subcircular-to-elliptical in *Centrosaurus* (AMNH FARB 5427[11]; Fig. 9i) and both juvenile and adult specimens of *Chasmosaurus* (CMN 2245, UALVP 52613[13,70]; Fig. 9m), and; very large (>100 mm), hexagonal-to-heptagonal, and characterized by a centrally-positioned or weakly off-center nipple-like structure in *Triceratops* (HMNS PV.1506; Larson et al.[20], Bell and Hendrickx[45]; Fig. 9k). Unlike the truncated-cone or conical

feature scales of *Psittacosaurus* and other dinosaurs, such as the abelisaurid *Carnotaurus*[78], the nipple-like structure of *Triceratops*, which corresponds to an elevated volcano-like prominence (~1–3 cm in height), occupies only half of the feature scale surface, the rest of the feature scale being flat (Fig. 9k). In all of these taxa (*Psittacosaurus, Carnotaurus, Triceratops*), it is unlikely that the feature scale bore a spine or a "bristle"-like structure— similar to those seen on the tail of *Psittacosaurus*—although bristle-like projections are present on some scales in the early-branching neornithischian *Kulindadromeus*[48,49]. No discernible pattern in the arrangement of the feature scales can be observed in *Psittacosaurus* (SMF R 4970) or *Centrosaurus* (AMNH FARB 5427) given the preservation of only a single feature scale in the latter. The feature scales from the adult *Chasmosaurus* (CMN 2245) are, however, arranged in irregular, longitudinal rows and are spaced 5–10 cm apart[13] (Fig. 9m). Irregularly spaced feature scales are also present in the juvenile specimen of *Chasmosaurus*[70]. In the adult *Chasmosaurus*, the feature scales are delimited by wide and deep interstitial tissue (the "circumscribing groove" of Sternberg[13]), which is also seen on the single feature scale of *Centrosaurus*[11]. In *Chasmosaurus* (CMN 2245), the general arrangement of feature scales remains consistent over the large patch of preserved skin, but scale diameter decreases ventrally[13]. The polygonal feature scales of *Triceratops* also do not seem to form any particular pattern but, unlike *Chasmosaurus*, they are more regularly spaced (~15–20 cm) and less variable in size. The basement scales on the flank of ceratopsians form a mosaic of polygonal scales varying in size, shape and elongation. They are, however, typically pentagonal or hexagonal and delimited by deep interstitial tissues (Fig. 9i, k–n). The basement scales are flat or weakly convex in *Psittacosaurus, Centrosaurus* (Fig. 9i) and both juvenile and adult specimens of *Chasmosaurus* (Fig. 9l–n) whereas those of *Triceratops* are nearly flat-to-strongly convex and only slightly smaller than the feature scales (Fig. 9k). The basement scales of *Centrosaurus* and *Chasmosaurus* are significantly smaller (up to 10 and 25 mm in *Centrosaurus* and *Chasmosaurus*, respectively; Fig. 9e, m) than those of *Triceratops* which, with a diameter of up to 90 mm (Fig. 9k), has the largest basement scales among dinosaurs (Fig. 11). The number of basement scales delimiting the largest feature scales also varies from more than ten scales in *Chasmosaurus* (Fig. 9m) to typically seven or eight in *Triceratops* (Fig. 9k). In *Chasmosaurus*, basement scales associated with the feature scales are relatively large (10–25 mm) (Fig. 9m), although patches of smaller (3–5.5 mm) polygonal basement scales (Fig. 9n) are present elsewhere on the body[13].

Psittacosaurids are to our knowledge the only ceratopsians that preserve skin from the hindlimb and anterior portion of the tail, as well as details of the cloaca and ischial callosity (ref. [25] this paper). An indeterminate patch of skin described by Ji[21] in the possible psittacosaurid specimen GMC LL2001-01 is here interpreted as being from the tarsal region (Fig. 10b). As in SMF R 4970, the 7 cm$^2$ patch of skin consists of polygonal basement scales (2–3.5 mm) diminishing in size distally towards the metatarsals. Skin is also preserved in the tail region of *Psittacosaurus houi* (= *P. lujiatunensis*) PKUP V1051[79] but its integument was neither described nor illustrated in detail and it is unknown whether the tail of SMF R 4970 and PKUP V1051 shared the same scale morphology and arrangement. The *Centrosaurus* specimen TMP 1986.018.0097 is to our knowledge the only ceratopsid to preserve skin from the tail (Fig. 9j). Several patches of skin are preserved in the ventral and central regions of the distal portion of the tail and are made of irregular to polygonal basement scales (3–6 mm), some of which are diagonally oriented and anteroposteriorly elongated.

## Conclusions

The Frankfurt specimen of *Psittacosaurus* retains the highest percentage of body covering and best-preserved squamous skin of any dinosaur and is therefore central to the understanding of dinosaur appearance and biology. Although tail "bristles" in *Psittacosaurus* were described nearly two decades ago[16], LSF has afforded a far more detailed view of its integument, including new information on the anatomy and homology of the tail "bristles" as well as providing evidence for countershading in this taxon[19,23]. A thorough analysis of the specimen under LSF here reveals the full complexity and variation in the squamous integument of *Psittacosaurus*. Such complexity is in line with the emerging picture from other squamous-skinned ornithischians and saurischians, and which deviates from the over-simplified "scaly reptile" image of many dinosaurs[4,6,9,44,45,49,78,80]. Ironically, complexity in both architecture and function of epidermal scales is a commonality shared between dinosaurs and extant sauropsids (e.g. refs. [81,82]), although the potential functionality of various scale types is only now being explored in the former[44,45].

LSF provides remarkable resolution of the scales of SMF R 4970. In particular, new details are revealed regarding the keratinous jugal covering, the ischial callosity, feature scales on the mid-distal tail, and the arthral arrangement of the digital pads. The cloaca[25] is revealed here to have had a longitudinal vent, a similarity it shares only with modern crocodylians. This could imply similar cloacal anatomy in *Psittacosaurus* that combines a ventrally-positioned copulatory organ[63], and a decoupled ureter that empties into the coprodeum. Other crocodile-like integumentary features include quadrangular and transversely-banded abdominal and caudal scales on the ventral part of the animal. The jugal horn was apparently only covered dorsally by a sheet-like keratinous covering, which differs from earlier reconstructions (e.g. ref. [19]) but which was potentially variable between species and life stages.

Compared to other ornithischians (e.g., hadrosaurids), few ceratopsians are known to preserve skin and even fewer have been formally described. Nevertheless, some patterns are emerging. A hexagram arrangement of scales present on the limbs of *Psittacosaurus* and *Nasutoceratops* is, at this stage, a unique ceratopsian feature. In all ceratopsians where skin is preserved, the flank bears feature scales set into a matrix of smaller basement scales. However, *Triceratops* is unique in having polygonal feature scales that are only slightly larger than the basement scales, and which have a central, nipple-like protrusion. Interspecific differences in the architecture of both feature scales and basement scales support early comments on the taxonomic utility of scale patterns in Ceratopsia[13] and dinosaurs more broadly (e.g. refs. [2,6,9,10,83]).

## Materials and methods

The specimen SMF R 4970, referred to *Psittacosaurus* sp.[16,19,23] (Fig. 1), comes from the Early Cretaceous Jehol deposits of the Liaoning Province, China, and most likely from the Jianshangou Bed, Yixian Formation (126–130 Ma; Barremian/Aptian[84,85]) of the Sihetun locality, Beipiao County[16]. SMF R 4970 is on public display at the Senckenberg Research Institute and Natural History Museum, Frankfurt, Germany as part of the permanent exhibition and remains available for scientific study by qualified researchers.

Vinther et al.[19] (Supplementary Information) discussed the taphonomy of this specimen, which we follow here. Compression fossils, including SMF R 4970, do not exhibit any evidence of widening or distortion as a result of compression[19,86,87]. We therefore interpret the scales in SMF R 4970 as true, undistorted representations of the original keratinous integument (see also Vinther et al.[19], Supplementary Information). SMF R 4970 was photographed using LSF performed using an updated version of the methodology proposed by Kaye et al.[28] and refined in Wang et al.[29]. A 405 nm blue near-UV laser diode was used to fluoresce the specimen following standard laser safety protocol. Long exposure images were taken in a darkened room with a Nikon D810 DSLR camera fitted with a 425 nm blocking filter and controlled from a laptop using *digiCamControl*. Image post-processing (equalization, saturation and color balance) was performed uniformly across the entire field of view in *Photoshop CS6*. Because soft tissue

outlines in SMF R 4970 are best visible using LSF, the observations made using this technique and the resulting digital images formed the basis for the following descriptions. All measurements of the integument were taken using digital images uploaded and calibrated in ImageJ v1.52q. Average scale dimensions for each body region (Supplementary Table 1) were calculated from measurements of at least five randomly-selected scales per body region (see Supplementary Data).

The scaly skin of SMF R 4970 was compared to that of other dinosaurs, with a particular attention to ceratopsians (Supplementary Table 2), as well as snakes, lizards, birds, and crocodylians (see Supplementary Data). A representative sample of each of these extant groups were observed in the collections of the University of New England's Natural History Museum (Armidale, Australia) and photographed using an Olympus S7X7 stereomicroscope fitted with an Olympus SC50 digital camera (see Supplementary Data). Multifocal image stacks were manually captured using cellSens Standard (www.olympus-lifescience.com) imaging software and stacked in Adobe Photoshop CC 2019. Data on cloacal morphologies as well as horn dimensions were also gathered in various amniotes based on the literature or personal observations for comparison (see Supplementary Data).

**Age of the individual.** Bell et al.[24] recently assessed the age of SMF R 4970 based on femoral length. To summarize those findings, the right femur of SMF R 4970 is ~140 mm long, which is similar to the femoral lengths of the specimens of *P. houi* (a senior synonym of *P. lujiatunensis*; see Mayr et al.[23]) IVPP V12617 (138 mm) and V18344 (145 mm), LPM R00128 (135 mm) and R00138 (144 mm) and PKUP V1053 (149 mm) and V1056 (135 mm), which belong to ~6–7 year old subadults (see Table 1 and Fig. 5 of Erickson et al.[88] and Supplementary Table S2 of Zhao et al.[89]). This age is just shy of sexual maturity and at the beginning of the exponential growth phase (see Table 1 and Fig. 5 of Erickson et al.[88] and Supplementary Table S2 of Zhao et al.[89]). The femoral length of SMF R 4970 is the closest match to a nearly sexually mature subadult (see Table 1 and Fig. 5 of Erickson et al.[88] and Supplementary Table S2 of Zhao et al.[89]).

**Horn measurements and scale length vs. body mass.** The lengths of the bony core and keratinous "sheath" of the jugal horn of SMF R 4970 were measured following Brown's[38] method for the "spine length (SL)", i.e., from the base of the anterior margin of the bony core and keratinous "sheath" to its apex. To visualize the size of the basement scales compared to that of the body in dinosaurs, *Psittacosaurus* was plotted on an updated version of the graph published by Hendrickx and Bell[78] that plots the diameter of the largest basement scales against body mass. Our updated dataset includes 39 specimens of non-coelurosaurian dinosaurs, among which 34 genus taxa (see Supplementary Data). Information on the largest basement scales were mostly taken from the literature or Hendrickx and Bell[78] who additionally measured the largest basement scales on photos or figures using ImageJ in three taxa. They also mainly used the estimations of body mass published by Benson et al.[90] and applied the formulas provided by these authors to taxa absent from their dataset using limb bones length and circumference (see Supplementary Data).

**Terminology and taxonomy.** Scale terminology largely follows that outlined by Bell[9] and Hendrickx et al.[6]. The term "inclusion" (or scale inclusion) follows its use in crocodylian literature to refer to small, variably-shaped scales that fill irregular gaps between adjacent larger, more uniformly sized/shaped scales that form the main basement. In reference to the shape of the feature scales, "basal" is defined as toward the dermis (i.e., anatomically deep), whereas "apex/apical" is away from the dermis (anatomically superficial). *Psittacosaurus* taxonomy follows that of Sereno[31] and Hedrick and Dodson[33], although we acknowledge the differing opinions of other authors (e.g. ref.[91]).

**Reporting summary.** Further information on research design is available in the Nature Research Reporting Summary linked to this article.

## Data availability
All data generated or analyzed during this study are included in this published article (and its Supplementary Information files). These data are also available from the corresponding authors P.R.B. (pbell23@une.edu.au), C.H. (christophendrickx@gmail.com) and M.P. (mpittman@cuhk.edu.hk).

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

## Acknowledgements

We thank Rainer Brocke and Olaf Vogel (SMF) for specimen access; K. Vernes (UNE) for access to comparative collections; Jordan Mallon (CMN) for information on *Triceratops* integument, and; Caleb Brown (TMP), Carl Mehling (AMNH), Shyong En Pan (CMN), Scott Rufolo (CMN), Jordan Mallon (CMN), Nathan Enriquez (University of New England), David C. Evans (ROM), Eric K. Lund (NCM), Mark Loewen (NHMU), Marshal A. Fazio, Victoria Arbour (RBCM) and Albert Prieto-Marquez (Catalan Institute of Paleontology Miquel Crusafont) for sharing photos of the skin of *Centrosaurus*, *Chasmosaurus*, *Nasutoceratops*, *Triceratops*, *Liaoningosaurus* and *Edmontosaurus*. We especially thank Philip J. Currie (University of Alberta) for sharing numerous photos of the skin of *Triceratops* and the juvenile and adult individuals of *Chasmosaurus* as well as Caleb Brown for taking many photos of the skin of the *Centrosaurus* tail. Waylon Rowley is warmly thanked for sharing Lessem's article[76] from Discover. The authors additionally thank Eydie Rojas (HMNS) for providing the specimen number of the *Triceratops* skin at the HMNS, Guo Yu of the Geological Museum of China for information about specimen GMC LL2001-01, and Brontops (Twitter) for helpful discussions and bringing several important specimens to our attention. Helpful reviews by Andrew Farke, two anonymous reviewers and editorial comments from Luke Grinham and Katie Davis (handling editor) considerably improved the final version of this paper. C.H. is supported by the Consejo Nacional de Investigaciones Científicas y Técnicas (CONICET) and Agencia Nacional de Promoción Científica y Tecnológica, Argentina (Beca Pos-doctoral CONICET Legajo 181417). M.P. is supported by the School of Life Sciences, The Chinese University of Hong Kong (CUHK). T.G.K. is supported by the Foundation for Scientific Advancement. C.H. dedicates this work to Graciela Martin de Apud and Teresa Apud de Giuliano.

## Author contributions

Conceptualization: C.H., P.R.B. and M.P.; Methodology: P.R.B., C.H., M.P. and T.G.K.; Investigation: P.R.B., C.H. and M.P.; Resources: M.P., T.G.K. and G.M.; Writing—original draft: P.R.B. and C.H.; Writing—review and editing: P.R.B., C.H., M.P. and G.M.; Visualization: P.R.B., C.H., M.P. and T.G.K.; Funding acquisition: C.H., M.P. and T.G.K.

## Competing interests

The authors declare no competing interests.
