## [Peer Review File · Communications Biology]

Reviewers' comments:

Reviewer #1 (Remarks to the Author):

Bell et al. describes the integument of Psittacosaurus based on an exceptionally preserved specimen. I liked the paper and I suggest publication of the manuscript, after modification of the methods, discussion and figures.

Correlation between external and internal morphology of the cloaca: as the authors suggest in their discussion, a neat correlation between external and internal morphology is not necessary present among extant taxa (see squamates and birds). Therefore, would ask the authors to tone down their interpretations.

The methods say nothing about the graph presented in Figure X. How did the authors plot the graph between logscale length and body mass? From where did you get the body mass estimates? Please, expand the methods section, because it is not sufficient in the current state.

Figures: I am a little bit shocked by the quality of the drawings included in the figures. For example, Figure 1I is terrible to look at. The same applies to other skin drawings. I would strongly encourage the authors to include way more details in these. In their current form, they have no explanatory power.

Figure 10: please, take away the lines.

Reviewer #2 (Remarks to the Author):

This contribution by Bell and colleagues presents new data on skin in a specimen of Psittacosaurus, and synthesizes existing data on other ceratopsians to present a broad overview of the integument in this clade. The information contained herein is novel (in the case of new observations) or analyzed in a novel manner (in the case of previously published data). The conclusions, overall, seem consistent with the data, and the extensive descriptions, interpretations and illustrations will be very useful for future researchers. The audience will be not just ceratopsian researchers, but also anyone who is studying the evolution of integument in dinosaurs (a very hot topic).

I have only minor suggestions, outlined below.

1) The text largely accepts at face value that the morphologies presented here represent the life condition, especially for gross body shape. To what extent could aspects of soft tissue outline of body shape be affected by decomposition / bloating / sloughing of skin or compression of a 3D body into a more two-dimensional form?

2) Line 422-436: This section is **incredibly** speculative, and I think could be condensed considerably or removed without loss to the paper. Sheath vs. core length are highly variable (including and especially in the bovids cited here), and so I think it is very hazardous to use the "cheek horn" length of one Psittacosaurus specimen to infer anything about horn lengths in other ceratopsians.

3) line 586: The reference here to Protoceratops skin is almost certainly the specimen described by Brown & Schlaikjer 1940 (Plate 13), in their monograph on the taxon. You clarify this later (line 618/619), so the text should be consistent throughout. My own opinion as well as general consensus amongst people I've talked with (I don't think this has been published) is that this was probably not preserving fine-level skin, but a diagenetic crust. Note that there are areas covered with this rind that don't make sense as skin -- e.g., the gap between upper and lower jaws, external naris, etc. I'm skeptical that we can tell anything about skin texture from this specimen -- you potentially see pebbly texture on the matrix separated from the skeleton, etc., and I am not sure I trust much from a fairly small photo in the original publication.

4) Jack Horner and colleagues have posited a keratinous covering over the skull in Triceratops

(akin to horn), including the frill. I don't think this was ever formally published (maybe in an SVP abstract?), but it is worth mentioning if you can track it down. (FWIW, I am skeptical of the fully sheathed skull claim, esp. in light of the material showing scales on the frill).

5) The colors/symbols in Figure 10 should be adjusted so that the image is more accessible for color-blind individuals; the reds, greens, and purples may be difficult to distinguish for some people.

--Andrew A. Farke (my name may be revealed to the authors)

Reviewer #3 (Remarks to the Author):

The manuscript presents a detailed description of the elements that form the external structure of the skin of a specimen of the ceratopsian dinosaur *Psittacosaurus* (SMF R 4970). The specimen is already known for its extremely exceptional preservation of the foot and some of the elements have already been described in previous works, but obtaining images by means of Laser-Stimulated Fluorescence reveals details of the structure of the skin of this animal unknown until now. time and are highly relevant. The analysis shows an unexpected diversity of the scale complex in these dinosaurs, provides information about the appearance of the animal in life (for example, in the color patterns, the morphology and size of the jugal horns or the distribution of the plantar pads), allows infer the probable structure of some soft structures (such as those of the cloaca) and provide information to interpret some characters in related dinosaurs.

The manuscript is well structured and provides very detailed descriptions of the structures considered. Comparisons with the skin structures of other ornithichians are very exhaustive, including specimens poorly documented up to now. Some inferences can be relatively speculative (such as some aspects of interpreting details of the anatomy of the cloaca from its general resemblance to that of crocodylians, or interpreting the size of many horns of different ceratopsians from the jugal horn) but The text clearly explains the scope of these interpretations.

Given the documentary and descriptive nature of a large part of the manuscript, the figures are very relevant and those provided are of high quality and effectively support the proposed text. The manuscript is well presented, developed and justified. Probably some numerical analysis or geometric morphometry of the scales (more elaborate than those presented in figure 10 and scarcely discussed in the text) can be developed later, but the descriptive study of such an exceptional specimen, enhanced by a novel analysis technique that it effectively amplifies the information, and the direct comparison presented is relevant enough to advise publication of the manuscript in its current state.

The 30th of May 2022

Rebuttal letter

We here re-submit our manuscript entitled “*The Exquisitely Preserved Integument of Psittacosaurus and the Scaly Skin of Ceratopsian Dinosaurs*” to *Communications Biology*. The comments, remarks, suggestions and corrections from the three reviewers were all taken into consideration and incorporated in the new version of the MS. We appreciate the positive remarks provided by the reviewers and agree with almost everything they highlighted to improve the quality of the manuscript. We only disagree with reviewer 1 on the quality of the drawing. We indeed believe that it is unnecessary to include more detail (such as a drawing of every single scales) as the figures already provide the necessary information to fully understand the morphology of the *Psittacosaurus* integument. We have, however, updated Fig. 1 for clarity reasons.

The main changes in our revised MS results from: i) the inclusion of the new section “*Horn measurements and scale length vs. body mass*” in the Material and Methods to provide the necessary data that explains the graph illustrated in Fig. 11; ii) the deletion of most of the text on horn length estimations in ceratopsids using the proportion between the jugal horn core and keratinous ‘sheath’ in *Psittacosaurus*; and iii) the inclusion of three additional ceratopsian specimens with skin (i.e., a possible psittacosaurid reported by Ji (2004) and skin patches from the *Centrosaurus* tail and an indeterminate body part of *Chasmosaurus*) leading to the creation of a new figure (Fig. 9) and the revision of Fig. 10. Please find below our responses in green, which follow each specific comment from the referees.

Sincerely,

Dr. Christophe Hendrickx, San Miguel de Tucumán, Argentina

Reviewers' comments:

Reviewer #1 (Remarks to the Author):

Bell et al. describes the integument of *Psittacosaurus* based on an exceptionally preserved specimen. I liked the paper and I suggest publication of the manuscript, after modification of the methods, discussion and figures. **We thank the reviewer for the positive review and helpful comments.**

Correlation between external and internal morphology of the cloaca: as the authors suggest in their discussion, a neat correlation between external and internal morphology is not necessary present among extant taxa (see squamates and birds). Therefore, would ask the authors to tone down their interpretations. **Our wording regarding cloacal form in squamates was potentially misleading where we stated “the condition in lizards is highly variable...”. By this we mean, the *scale pattern* is variable, but *not* the cloacal form (transverse) or internal anatomy. We have changed the text accordingly. Although there is some variation in birds, these differences (in internal anatomy) are phylogenetically relevant: ‘primitive’ birds (palaeognaths) share the same internal configuration with crocodylians. More importantly, these two groups phylogenetically bracket dinosaurs. Our arguments for the condition in *Psittacosaurus* are based on both direct anatomical evidence and the extant phylogenetic bracket as explicitly stated in the original manuscript. Nevertheless, we have been careful to use cautious wording when extrapolating the internal anatomy of a dinosaur from modern exemplars. For example, we state “the longitudinal vent of *Psittacosaurus* potentially implies a crocodylian-like internal anatomy” and “we hypothesise the presence of...”. We therefore believe that we have used a suitable tone in our language that does not ‘force’ our interpretation but does, however, provide a sound basis for advancing our knowledge of a novel aspect of dinosaurian soft tissue anatomy. We do acknowledge that this wording was not so cautious in the Conclusion and we have modified the following sentence to: “a similarity it shares only with modern crocodylians and which could imply similar cloacal anatomy in *Psittacosaurus*”.**

The methods say nothing about the graph presented in Figure X. How did the authors plot the graph between logscale length and body mass? From where did you get the body mass estimates? Please, expand the methods section, because it is not sufficient in the current state. **These have been added to the methods.**

Figures: I am a little bit shocked by the quality of the drawings included in the figures. For example, Figure II is terrible to look at. The same applies to other skin drawings. I would strongly encourage the authors to include way more details in these. In their current form, they have no explanatory power. **We believe that all the drawings associated with the figures provide the necessary information to fully understand the morphology of the *Psittacosaurus* integument. We do not consider that drawing every single scale is helpful to the reader. Likewise, this issue was not raised by the two other reviewers and we, therefore, left most of the figures untouched. Figure II was the only one to be revised to show the distribution of the different integumentary types in a slightly better way.**

Figure 10: please, take away the lines. **Done.**

Reviewer #2 (Remarks to the Author):

This contribution by Bell and colleagues presents new data on skin in a specimen of *Psittacosaurus*, and synthesizes existing data on other ceratopsians to present a broad overview of the integument in this clade. The information contained herein is novel (in the case of new observations) or analyzed in a novel manner (in the case of previously published data). The conclusions, overall, seem consistent with the data, and the extensive descriptions, interpretations and illustrations will be very useful for future researchers. The audience will be not just ceratopsian researchers, but also anyone who is studying the evolution of integument in dinosaurs (a very hot topic). I have only minor suggestions, outlined below. **We thank the reviewer for their helpful comments and time and effort in reviewing our manuscript.**

1) The text largely accepts at face value that the morphologies presented here represent the life condition, especially for gross body shape. To what extent could aspects of soft tissue outline of body shape be affected by decomposition / bloating / sloughing of skin or compression of a 3D body into a more two-dimensional form? **The taphonomy of this specimen was discussed by Vinther et al. (2016; sup. inf.) who also provide details on the high-fidelity preservation (microbodies, biomarkers etc) that indicate the skin is “true to form” and not distorted as a result of compaction, bloating etc. Importantly, this specimen is a compression fossil: aktuo-experiments by Derek Briggs and colleagues have shown conclusively that compression fossils are not expanded or distorted versions of their original forms but accurate 2D versions of a 3D form. We have added a new paragraph to the methods that explain our rationale (based on the findings of Vinther et al. 2016) for interpreting the scales “at face value”.**

2) Line 422-436: This section is **incredibly** speculative, and I think could be condensed considerably or removed without loss to the paper. Sheath vs. core length are highly variable (including and especially in the bovids cited here), and so I think it is very hazardous to use the “cheek horn” length of one *Psittacosaurus* specimen to infer anything about horn lengths in other ceratopsians. **We agree with the reviewer and have removed most of the size estimates for other ceratopsids. We have added other sentences that comment on the variability in bovids and the subsequent ‘hazard’ in applying the ratio seen in *Psittacosaurus* to other dinosaurs.**

3) line 586: The reference here to *Protoceratops* skin is almost certainly the specimen described by Brown & Schlaikjer 1940 (Plate 13), in their monograph on the taxon. You clarify this later (line 618/619), so the text should be consistent throughout. My own opinion as well as general consensus amongst people I've talked with (I don't think this has been published) is that this was probably not preserving fine-level skin, but a diagenetic crust. Note that there are areas covered with this rind that don't make sense as skin -- e.g., the gap between upper and lower jaws, external naris, etc. I'm skeptical that we can tell anything about skin texture from this specimen -- you potentially see pebbly texture on the matrix separated from the skeleton, etc., and I am not sure I trust much from a fairly small photo in the original publication. **We have added “Brown & Schlaikjer 1940 (Plate 13)” to each mention of *Protoceratops* skin as suggested as well as noted that this skin is only tentatively identified in each mention. We thank the reviewer for their insights into this particular *Protoceratops* specimen which has proven enigmatic. We have tempered our comments later in the comparisons and rather than overinterpreting the old photograph, have simply commented that “the presumed integument has since been entirely prepared off the specimen and verification of any of**

these interpretations is no longer possible.”

4) Jack Horner and colleagues have posited a keratinous covering over the skull in *Triceratops* (akin to horn), including the frill. I don't think this was ever formally published (maybe in an SVP abstract?), but it is worth mentioning if you can track it down. (FWIW, I am skeptical of the fully sheathed skull claim, esp. in light of the material showing scales on the frill). This hypothesis has indeed been published in a 2002 SVP abstract by Horner and Marshall (“Keratinous covered dinosaur skulls”). Our MS has, consequently, been updated accordingly, and now includes the following sentence: “A small piece of skin associated with the frill of the *Triceratops* specimen CMNFV 56508 (J. Mallon, pers. comm. 2021) reveals that the frill of this taxon, and probably all ceratopsians, were covered with polygonal basement scales, rejecting Horner and Marshall's (2002) hypothesis that a keratinous sheath covered nearly the entire skull of ceratopsians such as *Triceratops* and *Torosaurus*.”

5) The colors/symbols in Figure 10 should be adjusted so that the image is more accessible for color-blind individuals; the reds, greens, and purples may be difficult to distinguish for some people. The creator of Figure 10 (C.H.) is actually colour-blind and can, therefore, confirm that there is no issue with the colour scheme in this figure. Nonetheless, the figure has been updated to be read by people who are totally colour deficient.

--Andrew A. Farke (my name may be revealed to the authors)

Reviewer #3 (Remarks to the Author):

The manuscript presents a detailed description of the elements that form the external structure of the skin of a specimen of the ceratopsian dinosaur *Psittacosaurus* (SMF R 4970). The specimen is already known for its extremely exceptional preservation of the foot and some of the elements have already been described in previous works, but obtaining images by means of Laser-Stimulated Fluorescence reveals details of the structure of the skin of this animal unknown until now. time and are highly relevant. The analysis shows an unexpected diversity of the scale complex in these dinosaurs, provides information about the appearance of the animal in life (for example, in the color patterns, the morphology and size of the jugal horns or the distribution of the plantar pads), allows infer the probable structure of some soft structures (such as those of the cloaca) and provide information to interpret some characters in related dinosaurs.

The manuscript is well structured and provides very detailed descriptions of the structures considered. Comparisons with the skin structures of other ornithichians are very exhaustive, including specimens poorly documented up to now. Some inferences can be relatively speculative (such as some aspects of interpreting details of the anatomy of the cloaca from its general resemblance to that of crocodilians, or interpreting the size of many horns of different ceratopsians from the jugal horn) but The text clearly explains the scope of these interpretations. Given the documentary and descriptive nature of a large part of the manuscript, the figures are very relevant and those provided are of high quality and effectively support the proposed text. The manuscript is well presented, developed and justified. Probably some numerical analysis or geometric morphometry of the scales (more elaborate than those presented in figure 10 and scarcely discussed in the text) can be developed later, but the descriptive study of such an exceptional

specimen, enhanced by a novel analysis technique that it effectively amplifies the information, and the direct comparison presented is relevant enough to advise publication of the manuscript in its current state. **We thank reviewer 3 for their time and positive review of the manuscript.**